

# Estimation of Granger causality through Artificial Neural Networks: applications to physiological systems and chaotic electronic oscillators

Yuri Antonacci[1,2,3], Ludovico Minati[4,5], Luca Faes[6], Riccardo Pernice[6], Giandomenico Nollo[7], Jlenia Toppi[2,3], Antonio Pietrabissa[3] and Laura Astolfi[2,3]

[1] Department of Physics and Chemistry "Emilio Segrè", University of Palermo, Palermo, Italy
[2] Istituto di Ricovero e Cura a Carattere Scientifico (IRCCS) Fondazione Santa Lucia, Rome, Italy
[3] Department of Computer, Control and Management Engineering "Antonio Ruberti", University of Rome "La Sapienza", Rome, Italy
[4] Center for Mind/Brain Sciences (CIMeC), University of Trento, Trento, Italy
[5] Institute of Innovative Research, Tokyo Institute of Technology, Yokohama, Japan
[6] Department of Engineering, University of Palermo, Palermo, Italy
[7] Department of Industrial Engineering, University of Trento, Trento, Italy

Corresponding author
Yuri Antonacci,
yuri.antonacci@uniroma1.it

## ABSTRACT

One of the most challenging problems in the study of complex dynamical systems is to find the statistical interdependencies among the system components. Granger causality (GC) represents one of the most employed approaches, based on modeling the system dynamics with a linear vector autoregressive (VAR) model and on evaluating the information flow between two processes in terms of prediction error variances. In its most advanced setting, GC analysis is performed through a state-space (SS) representation of the VAR model that allows to compute both conditional and unconditional forms of GC by solving only one regression problem. While this problem is typically solved through Ordinary Least Square (OLS) estimation, a viable alternative is to use Artificial Neural Networks (ANNs) implemented in a simple structure with one input and one output layer and trained in a way such that the weights matrix corresponds to the matrix of VAR parameters. In this work, we introduce an ANN combined with SS models for the computation of GC. The ANN is trained through the Stochastic Gradient Descent L1 (SGD-L1) algorithm, and a cumulative penalty inspired from penalized regression is applied to the network weights to encourage sparsity. Simulating networks of coupled Gaussian systems, we show how the combination of ANNs and SGD-L1 allows to mitigate the strong reduction in accuracy of OLS identification in settings of low ratio between number of time series points and of VAR parameters. We also report how the performances in GC estimation are influenced by the number of iterations of gradient descent and by the learning rate used for training the ANN. We recommend using some specific combinations for these parameters to optimize the performance of GC estimation. Then, the performances of ANN and OLS are compared in terms of GC magnitude and statistical significance to highlight the potential of the new approach to reconstruct causal coupling strength and network topology even in challenging conditions of data paucity. The results highlight the importance of of a proper selection of regularization parameter which determines the degree of sparsity in the estimated network. Furthermore, we apply the two approaches to real data scenarios,

to study the physiological network of brain and peripheral interactions in humans under different conditions of rest and mental stress, and the effects of the newly emerged concept of remote synchronization on the information exchanged in a ring of electronic oscillators. The results highlight how ANNs provide a mesoscopic description of the information exchanged in networks of multiple interacting physiological systems, preserving the most active causal interactions between cardiovascular, respiratory and brain systems. Moreover, ANNs can reconstruct the flow of directed information in a ring of oscillators whose statistical properties can be related to those of physiological networks.

# INTRODUCTION

A fundamental problem in the study of dynamical systems in many domains of science and engineering is to investigate the interactions among the individual system components whose activity is represented by different recorded time series. The evaluation of the direction and strength of these interactions is often carried out employing the statistical concept of causality introduced by *Wiener (1956)* and formalized in terms of linear regression analysis by *Granger (1969)*. Wiener–Granger causality (GC) was firstly introduced in the framework of linear bivariate autoregressive modeling in its unconditional form for which a generic time series $X$ is said to Granger-cause another series $Y$ if the past of $X$ contains information that helps to predict the future of $Y$ above and beyond the information already contained in the past of $Y$ (*Granger, 1969*). In the presence of more than two interacting system components, to take into account the presence of other time series which can potentially affect the two time series under analysis the bivariate formulation has been extended to the multivariate case through the use of vector autoregressive (VAR) models, leading to the computation of a conditional form of GC (*Geweke, 1984*). Due to its linear formulation, GC is very easy to implement, with very few parameters to be estimated if compared with model-free approaches and with a reduced computational cost (*Porta & Faes, 2015*).

GC from a driver to a target time series is typically quantified by comparing the prediction error variance obtained from two different linear regression models: (i) the "*full model*", in which the present sample of the target series is regressed on the past samples of all the time series in the dataset; (ii) the "*restricted model*", in which the present of the target is regressed on the past of all the time series excluding the driver (*Barnett & Seth, 2014*). However, this formulation does not take into account that, from a theoretical point of view, the order of the restricted model is infinite, leading to a strong bias or a very large variability associated with the estimation of GC, depending on the model order selected (*Stokes & Purdon, 2017*; *Faes, Stramaglia & Marinazzo, 2017*; *Barnett, Barrett &*

*Seth, 2018*). To overcome the latter problem, an approach based on state-space (SS) modeling of the observed VAR process has been introduced (*Barnett & Seth, 2015*); SS models provide a closed-form SS representation of the restricted VAR model and thus, starting from the identification of the full model only, GC in its conditional and unconditional form can be retrieved with high computational reliability directly from the SS parameters (*Solo, 2016*; *Barnett & Seth, 2015*; *Faes, Stramaglia & Marinazzo, 2017*).

The literature provides different methodologies for VAR model identification, such as the solution of the Yule-Walker equations through Levison's recursion or the Burg algorithm (*Kay, 1988*) by using the closed-form solution of Ordinary least squares (OLS) estimator, or more sophisticated such as those based on Artificial Neural Networks (ANNs). ANNs have become very popular in recent years, and they have been extensively used as a modeling tool because they are data-driven self-adaptive methods and can work as universal functional approximators (*Hornik, Stinchcombe & White, 1989*; *Hornik, 1991*). The ANN structure used for linear regression comprises one input layer and one output layer which are linked by a matrix of weights obtained after training the network. During the training process, the inputs are presented to the network and the weights are adjusted to minimize the distance between the real and predicted output using error backpropagation techniques (*Bishop, 1995*).

However, regardless of the methodology used to approach the regression problem, the estimation may be problematic in the setting of many observed processes and short time series available (*Antonacci et al., 2019a*). The literature reports that the stability and the existence of the solution for a linear regression problem are ensured when the number of data points is an order of magnitude greater than the number of VAR coefficients to be estimated (*Schlögl & Supp, 2006*; *Lütkepohl, 2013*). To cope with the issues arising in GC estimation when the ratio between data size and number of unknown parameters is low, different approaches have been proposed such as the use of time-ordered restricted VAR models (*Siggiridou & Kugiumtzis, 2015*), or the so-called partial conditioning (*Marinazzo, Pellicoro & Stramaglia, 2012*), and of penalized regression techniques based on the $l_1$-norm (LASSO regression) (*Antonacci et al., 2020b*; *Tibshirani, 1996*; *Pagnotta, Plomp & Pascucci, 2019*). In the latter case, the solution of the linear regression problem is found adding a constraint to the cost function to be minimized, usually the Mean Squared Error (MSE), that induces variable selection of the VAR parameters with a consequent reduction of the MSE associated with the estimation process. Based on $l_1$-constrained problems, in recent years, different $l_1$-regularized algorithms have been developed to avoiding overfitting during the training of ANNs. Moreover, the $l_1$-norm can be applied directly on the weights of the network during the training phase in an efficient way through stochastic gradient descent $l_1$ (SGD-$l_1$) (*Tsuruoka, Tsujii & Ananiadou, 2009*). While the use of ANNs as a VAR model for GC estimation has been proposed in both linear (*Talebi, Nasrabadi & Mohammad-Rezazadeh, 2018*) and non-linear frameworks (*Montalto et al., 2015*; *Attanasio & Triacca, 2011*; *Duggento, Guerrisi & Toschi, 2019*), the implementation of SGD-$l_1$ has never been tested for the purpose of reducing the effects of data paucity on the estimation of GC.

In the present work, an ANN used as a VAR model is embedded in the SS framework for the computation of GC (conditional and unconditional) and compared with the traditional OLS regression both in benchmark networks of simulated multivariate processes and in real-data scenarios. In simulations, we show how training parameters that are typically chosen in a heuristic way (i.e., learning rate and the number of iterations of gradient descent) can affect the estimation of GC in conditions of data paucity; after optimizing these parameters, we test the performance in the quantification of GC magnitude and statistical significance, reflecting respectively coupling strength and structure of the investigated directed functional network, comparatively with standard OLS identification. In real data analysis, we compare the two approaches first in physiological time series, reporting the evaluation of information flow and topology of the network of interactions between brain and peripheral systems probed in healthy subjects in different conditions of mental stress elicited by mental arithmetic and sustained attention tasks (*Antonacci et al., 2020b*; *Zanetti et al., 2019*), and then in signals produced by electronic circuits, showing how GC measures can describe the effect of remote synchronization previously observed in a ring of coupled chaotic oscillators (*Gambuzza et al., 2013*; *Minati, 2015a*; *Minati et al., 2018*).

The algorithms for the training of ANNs based on SGD-$l_1$ algorithm with the subsequent computation of GC by exploiting the SS framework are collected in the ANN-GC MATLAB toolbox, which can be downloaded from https://github.com/YuriAntonacci/ANN-GC-Toolbox.

## METHODS

### Vector autoregressive model identification

Let us consider a dynamical system $\mathcal{Y}$ whose activity is mapped by a discrete-time stationary vector stochastic process composed of $M$ real-valued zero-mean scalar processes, $\mathbf{Y} = [Y_1 \cdots Y_M]$. Considering the time step $n$ as the current time, the present and the past of the vector stochastic process are denoted as $\mathbf{Y}_n = [Y_{1,n} \cdots Y_{M,n}]$ and $\mathbf{Y}_n^- = [\mathbf{Y}_{n-1}\mathbf{Y}_{n-2}\cdots]$, respectively. Moreover, assuming that $\mathbf{Y}$ is a Markov process of order $p$, its whole past history can be truncated using $p$ time steps, i.e., using the $Mp$-dimensional vector $\mathbf{Y}_n^p$ such that $\mathbf{Y}_n^- \approx \mathbf{Y}_n^p = [\mathbf{Y}_{n-1}\cdots\mathbf{Y}_{n-p}]$. Then, in the linear signal processing framework, the dynamics of $Y$ can be described by the vector autoregressive (VAR) model:

$$\mathbf{Y}_n = \sum_{k=1}^{p} \mathbf{Y}_{n-k}A_k + \mathbf{U}_n, \tag{1}$$

where $A_k$ is an $M \times M$ matrix containing the VAR coefficients, and $\boldsymbol{U} = [U_1 \cdots U_M]$ is a vector *of $M$ zero-mean white processes, denoted as innovations, with $M \times M$ covariance* matrix $\sum \equiv \mathbb{E}[U_n^T U_n]$ ($\mathbb{E}$ is the expected value).

Let us now consider a realization of the process $\mathbf{Y}$ involving $N$ consecutive time steps, collected in the $N \times M$ data matrix $[\mathbf{y}_1;\cdots;\mathbf{y}_N]$, where the delimiter ";" stands for row separation, so that the $i^{th}$ row is a realization of $\mathbf{Y}_i$, i.e., $\mathbf{y}_i = [y_{1,i}\ldots y_{M,i}]$, $i = 1,\ldots,N$, and the

$j^{th}$ column is the time series collecting all realizations of $Y_j$, i.e., $[y_{j,1}...y_{j,N}]^T$, $j = 1,...,M$, . The Ordinary least squares (OLS) identification finds an optimal solution for the problem (1) by solving the following linear quadratic problem:

$$\widehat{A} = \text{argmin}_A ||y - y^p A||_2^2, \tag{2}$$

where $y = [\mathbf{y}_{p+1};\cdots;\mathbf{y}_N]$ is the $(N-p) \times M$ matrix of the predicted values, $y^p = [\mathbf{y}_{p+1}^p;\cdots;\mathbf{y}_N^p]$ is the $(N-p) \times Mp$ matrix of the regressors and $A = [A_1;\cdots;A_p]$ is the $Mp \times M$ coefficient matrix. The problem has a solution in a closed form $\widehat{A} = ([y^p]^T y^p)^{-1} [y^p]^T y$ for which the residual sum of squares (RSS) is minimized (*Lütkepohl, 2013*).

## Artificial neural networks as a vector autoregressive model

Let consider a generic ANN described by the function $y = f(\mathbf{w};\mathbf{x})$ which takes as input a vector $\mathbf{x} \in R^d$ and outputs a scalar value $y \in R$;. In the following, we consider networks with a single output for the sake of simplicity, but all the treatments can be extended to the case of multiple outputs. The output of the network depends on a set of $Q$ adaptable parameters (i.e., the weights connecting the layers), that are collected in a single vector $\mathbf{w} \in R^Q$ to be optimized during the training process.

Given a training data set of $N$ input/output pairs $S = \{\mathbf{x}_i,y_i\}$, the learning task aims at solving the following regularized optimization problem:

$$\widehat{w} = \text{argmin}_w \frac{1}{N}\sum_{i=1}^{N} l(y_i,f(\mathbf{w};\mathbf{x}_i)) + \lambda r(\mathbf{w}), \tag{3}$$

where $l(\cdot,\cdot)$ is a convex function $\in C^1$, i.e, continuously differentiable with respect to $\mathbf{w}$, while $r(\cdot)$ is a convex regularization term with a regularization parameter $\lambda \in R^+$. A typical loss function used for the linear regression problem is the squared error of the regression analysis. Inspired by the LASSO algorithm, a way to enforce sparsity in the vector of weights is to penalize the cumulative absolute magnitude of the weights by using the $l_1$ norm as regularization term:

$$r(\mathbf{w}) = ||\mathbf{w}||_1 = \sum_{k=1}^{Q} |\mathbf{w}_k|. \tag{4}$$

Then, a possible way to solve the problem (3) is to use Stochastic Gradient Descent (SGD) that exploits a small randomly-selected subset of the training samples to approximate the gradient of the objective function. The number of training samples used for this approximation is the batch size. In the present work, we adopt a full batch approach in which all samples are considered, so that SGD simply translates into gradient descent. For each training sample $i$, the network weights are updated as follows:

$$\mathbf{w}^{j+1} = \mathbf{w}^j + \eta_j \frac{\partial}{\partial \mathbf{w}}(l(y_i,f(\mathbf{w};\mathbf{x}_i)) - \frac{\lambda}{N}\sum_{k=1}^{Q} |\mathbf{w}_k|), \tag{5}$$

where $j$ is the iteration counter and $\eta_j$ is the learning rate at each iteration. The difficulty with $l_1$ regularization is that the last term on the right-hand side in (5) is not differentiable when the weight is zero. To solve this issue, following the procedure introduced in *Tsuruoka, Tsujii & Ananiadou (2009)* $l_1$ regularization with cumulative penalty is applied directly on the weights of the network during the training process.

Let $u_j$ be the absolute value of the total $l_1$ penalty received by each weight. Since the absolute value of the $l_1$ penalty does not depend on the weight and on the regularization parameter $\lambda$, it is the same for all the weights and is simply accumulated as:

$$u_j = \frac{\lambda}{N} \sum_{t=1}^{j} \eta_t. \tag{6}$$

At each training sample $i$, the weights of the network are updated as follows:

$$w_k^{j+\frac{1}{2}} = w_k^j + \eta_j \frac{\partial l(y_i, f(\mathbf{w}; \mathbf{x}_i))}{\partial \mathbf{w}} \Big|_{\mathbf{w}=\mathbf{w}^j}, \tag{7}$$

$$\textbf{if} \ \ w_k^{j+\frac{1}{2}} > 0 \ \ \textbf{then} \ w_k^{j+1} = max(0, w_k^{j+\frac{1}{2}} - (u_k + q_k^{j-1})), \tag{8}$$

$$\textbf{else if} \ \ w_k^{j+\frac{1}{2}} < 0 \ \ \textbf{then} \ w_k^{j+1} = min(0, w_k^{j+\frac{1}{2}} - (u_k - q_k^{j-1})), \tag{9}$$

where $q_k^j$ is the total $l_1$-penalty that $w_k$ has actually received:

$$q_k^j = \sum_{t=1}^{j} (w_k^{t+1} - w_k^{t+\frac{1}{2}}). \tag{10}$$

This method for updating the weights penalizes the weight according to the difference between $u_j$ and $q_k^{j-1}$ and is called SGD-$l_1$.

Generalizing the whole procedure to a network with multiple outputs, in the linear signal processing framework the optimization problem (3) can be solved by using a linear function $f(\cdot; \cdot)$ linking the input layer with the output layer. In particular, the structure of the neural network necessary for solving the regularized problem (3) in the linear framework is reported in Fig. 1 for the $n^{th}$ training sample. The input layer shows $Mp$ neurons representing the past history of the considered stochastic process, truncated at $p$ lags ($\mathbf{Y}_n^p$). The output layer is composed of $M$ neurons representing the present state of the whole system ($\mathbf{Y}_n$). The $Mp \times M$ matrix $\mathbf{W}$ contains the weights of the networks that describe the relationships existent between the output and the input layer. Considering all the $(N - p)$ training samples, the loss function $l(\cdot, \cdot)$ becomes:

$$l(\mathbf{y}, \mathbf{y}^p \mathbf{W}) = ||\mathbf{y} - \mathbf{y}^p \mathbf{W}||_2^2, \tag{11}$$

which highlights that the weight $\mathbf{W}$ corresponds to the matrix $\mathbf{A}$ containing the parameters of the VAR model (1). Thus, the described ANN is completely equivalent to a VAR model, except for the fact that the training process induces sparsity into the weight matrix $\mathbf{W}$. A feed-forward neural network with no hidden layers, like the one described above, is a generalized linear model that can be identified with an equivalent least squares

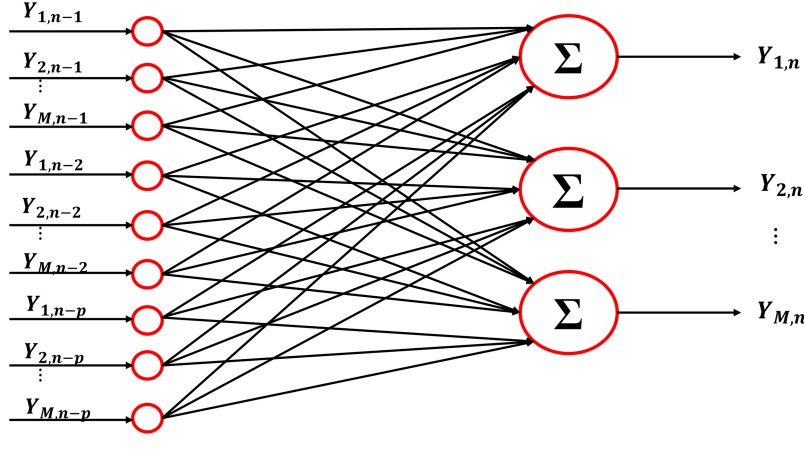

Input Layer                    Output Layer

**Figure 1 Schematic representation of the architecture of the Neural Network used as VAR model.** The input and the output of the network are represented by the lagged variables and by the present states of all processes included in the analysis.     

optimization problem with $l_1$ regularization applied to the estimated coefficients. If this regularization is not applied, and by using the loss function (11), the problem stated in (3) is completely equivalent to an OLS regression (*Sun, 2000*).

### Determination of the regularization parameter

The determination of the regularization parameter $\lambda$ is a key element of the estimation process, as its selection strongly influences the performance of resulting regression. For a high value of $\lambda$, the SGD-$l_1$ algorithm provides a matrix of weights **W** in which all entries are zero. On the other hand, when $\lambda \to 0$, the weights stored in **W** are all different from zero and the solution corresponds to the OLS solution (*Tibshirani, 1996*). In this work, the optimal value for $\lambda$ has been tested in the range $[\lambda_l, \lambda_u]$, where $\lambda_l$ and $\lambda_u$ are the values leading to maximum density (no zero elements) and maximum sparseness (all zero elements) of the weight matrix. Subsequently, following the procedure described in *Sun et al. (2016)*, with a hold out approach, we independently draw 90% of the samples available (rows of **y** and $\mathbf{y}^p$) as the training set and kept the remaining 10% for testing. Training and test sets were then normalized and, for each assigned $\lambda$, the number of non-zero weights was counted in the matrix $\widehat{\mathbf{W}}$ estimated on the training set, and the RSS was computed on the test set as well. This procedure was iterated for each $\lambda$, and the optimal $\lambda$ was taken as the value minimizing the ratio between RSS and the number of non-zero weights (*Sun et al., 2016*; *Antonacci et al., 2020b*; *Tibshirani & Taylor, 2012*). The weight matrix **W** obtained with the selected optimal $\lambda$ was then used for the subsequent GC analysis.

## Measuring Granger causality

Given the vector process $\mathbf{Y} = [Y_1 \cdots Y_M]$, let us assume $Y_j$ as the *target* process and $Y_i$ as the *source* process, with the remaining $M - 2$ processes collected in the vector $\mathbf{Y}_s$ where

$s = \{1,\ldots,M\}\backslash\{i,j\}$. Considering the past of the source process $Y_{i,n}^p$ and the past of the target process $Y_{j,n}^p$ we state that the $i^{th}$ process G-causes the $j^{th}$ process (conditional on the other $s$ processes), if $Y_{i,n}^p$ conveys information about $Y_{j,n}$ above and beyond the information contained in $Y_{j,n}^p$ and in all other processes $\mathbf{Y}_{s,n}^p$. This definition is implemented regressing the present of the target on the past of all processes (full regression) and on the past of all processes except the driver (restricted regression), to yield respectively the prediction errors $E_{j|ijs,n} = Y_{j,n} - \mathbb{E}[Y_{j,n}|\mathbf{Y}_n^p]$ and $E_{j|js,n} = Y_{j,n} - \mathbb{E}[Y_{j,n}|Y_{j,n}^p, \mathbf{Y}_s^p]$. The resulting prediction error variances, $\lambda_{j|ijs} = \mathbb{E}[E_{j|ijs,n}^2]$ and $\lambda_{j|js} = \mathbb{E}[E_{j|js,n}^2]$ are then combined to obtain the definition of GC (in its conditional form) from $Y_i$ to $Y_j$ (*Geweke, 1982*):

$$F_{i \to j|s} = \ln \frac{\lambda_{j|js}}{\lambda_{j|ijs}}. \tag{12}$$

Following a similar reasoning, the GC in its original form (unconditional) from $Y_i$ to $Y_j$ is defined as (*Granger, 1969*):

$$F_{i \to j} = \ln \frac{\lambda_{j|j}}{\lambda_{j|ij}}, \tag{13}$$

where $\lambda_{j|j} = \mathbb{E}[E_{j|j,n}^2]$ and $\lambda_{j|ij} = \mathbb{E}[E_{j|ij,n}^2]$ are the prediction error variances of the linear regression of $Y_{j,n}$ on $Y_{j,n}^p$ and on $[Y_{j,n}^p Y_{i,n}^p]$, respectively obtained from the errors $E_{j|j,n} = Y_{j,n} - \mathbb{E}[Y_{j,n}|Y_{j,n}^p]$ and $E_{j|ij,n} = Y_{j,n} - \mathbb{E}[Y_{j,n}|Y_{j,n}^p, Y_{i,n}^p]$.

The prediction error variances needed for the determination of the GC measures can be computed from the identification of the model (1) or by the training of the presented neural network, i.e., from the parameters $(\mathbf{A}_1,\ldots,\mathbf{A}_p,\sigma)$ estimated using OLS or from the weights $(\mathbf{W},\sigma)$ estimated through the SGD-$l_1$ training algorithm. Given that $E_{j|ijs,n} = U_{j,n}$, the error variance of the full regression can be obtained as the $j^{th}$ diagonal element of the error covariance matrix $\lambda_{j|ijs} = \sigma(j,j)$. The other partial variances in (12) and (13) can be retrieved, starting from the identification of the full model, by exploiting the theory of State-Space (SS) models (*Barnett & Seth, 2015*; *Faes, Marinazzo & Stramaglia, 2017*), according to which the VAR model (1) can be represented as an SS model relating the observed process $\mathbf{Y}$ to an unobserved process $\mathbf{Z}$ through the equations (*Barnett & Seth, 2015*; *Solo, 2016*):

$$\mathbf{Z}_{n+1} = \mathbf{Z}_n \mathbf{A} + \mathbf{E}_n \mathbf{K}, \tag{14}$$
$$\mathbf{Y}_n = \mathbf{Z}_n \mathbf{C} + \mathbf{E}_n, \tag{15}$$

where the innovations $\mathbf{E}_n = \mathbf{Y}_n - \mathbb{E}[\mathbf{Y}_n|\mathbf{Y}_n^p]$ are equivalent to the innovations $\mathbf{U}_n$ in (1) and thus have covariance matrix $\Phi = \mathbb{E}[\mathbf{E}_n^T \mathbf{E}_n] = \Sigma$. This representation, typically denoted as "innovation form" SS model (ISS) (*Barnett & Seth, 2015*), also evidences the Kalman Gain matrix $\mathbf{K}$, the state matrix $\mathbf{A}$ and the observation matrix $\mathbf{C}$, which can all be computed from the original VAR parameters in (1) as reported in (*Faes, Marinazzo & Stramaglia, 2017*). The advantage of this representation is that it allows to form "submodels" which exclude one or more scalar processes from the observation Eq. (15) leaving the state Eq. (14) unaltered. In particular, the submodels excluding the driver process $Y_i$, the group

of $s$ processes $\mathbf{Y}_s$, or the the driver process $Y_i$ and the group of $s$ processes $\mathbf{Y}_s$, have the following observation equations:

$$\mathbf{Y}_{js,n} = \mathbf{Z}_n \mathbf{C}^{(js)} + \mathbf{E}_{js,n}, \tag{16}$$

$$\mathbf{Y}_{ji,n} = \mathbf{Z}_n \mathbf{C}^{(ji)} + \mathbf{E}_{ji,n}, \tag{17}$$

$$\mathbf{Y}_{j,n} = \mathbf{Z}_n \mathbf{C}^{(j)} + \mathbf{E}_{j,n}, \tag{18}$$

where the superscripts $(js)$, $(ji)$ and $(j)$ denote the selection of the columns with indices $(js)$, $(ji)$ and $(j)$ in a matrix. As shown by (*Barnett & Seth, 2015*), the submodels (14,16), (14,17) and (14,18) are not in ISS form, but can be converted into ISS by solving a Discrete Algebraic Riccati equation (DARE). Then, the covariance matrices of the innovations $\mathbf{E}_{js,n}$, $\mathbf{E}_{ji,n}$ and $\mathbf{E}_{j,n}$ include the desired error variances $\lambda_{j|js}$, $\lambda_{j|ji}$ and $\lambda_{j|j}$ as the first diagonal element.

In order to establish the existence of a direct link from the $i^{th}$ node to the $j^{th}$ node of the network represented by the observed vector process, the statistical significance of the conditional GC computed after OLS identification of the VAR model was tested using surrogate data. Specifically, one hundred sets of surrogate times series were first generated using the Iterative Amplitude Adjusted Fourier Transform (IAAFT) procedure (*Schreiber & Schmitz, 1996*); then, for each directed link ($i,j$ pair), the conditional GC $F_{i \to j|s}$ was estimated for each surrogate set, a threshold equal to the $95^{th}$ percentile of its distribution on the surrogates was determined, and the link was considered as statistically significant when the estimated $F_{i \to j|s}$ was above the threshold. In the case of ANN identification, the statistical significance of the estimated conditional GC values was determined in a straightforward way exploiting the sparseness of the weights matrix $\mathbf{W}$ resulting from the training through SGD-$l_1$.

## Simulation experiments

This section reports three simulations designed to evaluate the performances of the proposed estimator of the GC based on ANNs trained with SGD-$l_1$ in comparison with the traditional VAR identification based on OLS. The first simulation evaluates the conditional GC computed by the ANN estimator in known structures of networks assessed with different amount of data samples, for different values of learning rate ($\eta$) and for different values of iterations of the SGD-$l_1$ algorithm. blueIn the second and in the third simulation studies, after having extracted the best combination of learning rate and the number of iterations of the gradient descent to be used in ANN-based estimation, we compare it with OLS estimation as regards the ability to retrieve the true values of the conditional GC and to reconstruct the assigned network topology. The effects of different values of signal-to-noise ratio (SNR) and of simulating a denser network structure are evaluated respectively in the second and in the third study. In all simulations, the topology is representative of the interaction of a ten-variate VAR process exhibiting a random interaction structure with two different values of density of connected nodes (*Toppi et al., 2016a*; *Antonacci et al., 2020b*; *Pascucci, Rubega & Plomp, 2020*).

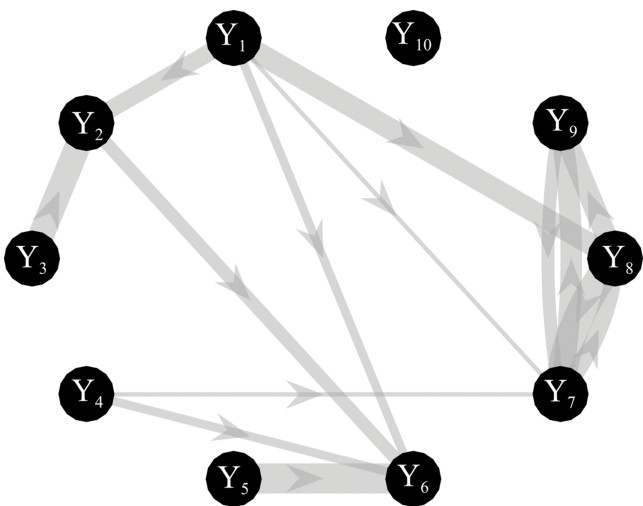

**Figure 2 Graphical representation of one of the ground-truth networks of the simulation study.**
Arrows represent the causal links randomly assigned between two network nodes via nonzero VAR
coefficients. The thickness of each arrow is proportional to the strength of the causal connection assessed
by the conditional GC, with minimum and maximum values equal to 0.0069 and 0.4. The number of
connections for each network is set to 14 out of 90.

### Simulation studies I-II

Simulated multivariate time series ($M = 10$) were generated as a realization of a VAR(16)
model fed by zero-mean independent Gaussian noise with variance equal to 0.1. The
simulated networks have a ground-truth structure with a density of connected nodes equal
to 15%, where non-zero AR parameters of values chosen randomly in the interval
$[-0.8, 0.8]$ were set at lags assigned randomly in the range (1–16) (*Anzolin & Astolfi, 2018*).
The knowledge of the true AR parameters allows computing the theoretical values of the
conditional GC and the true network topology, as illustrated for an exemplary case in
Fig. 2. Simulations were generated for different values of: (1) the parameter $K$ defined as
the ratio between the number of data samples available ($N \times M$) and the number of AR
coefficients to be estimated ($M^2 \times p$); (2) the signal-to-noise ratio (SNR) defined as the
ratio between the squared amplitude of the signal and the square amplitude of additive
white noise. One hundred networks were generated for each value of $K$ in the range
(1,3,10,20); the length of the simulated time series was $N = 160$ when $K = 1$ and $N = 3,200$
when $K = 20$. When additive noise was considered in the simulation study, SNR varies in
the range (0.1, 1, 5, 10, $10^3$).

First, considering ANN estimation performed for each value assigned to $K$ and for each
realization, the learning rate $\eta$ and the number of iterations for the SGD-$l_1$ during the
training process were varied respectively in the range ($10^{-3}$,$10^{-4}$,$10^{-5}$) and in the range
(100,1000,2000). Importantly, for each network structure a different neural network was
trained initializing the weights according to the method described in *Glorot & Bengio
(2010)* that guarantees a faster convergence of the gradient descent algorithm. After
training, the conditional GC between each pair of processes was estimated from the matrix
of the weights **W** using the SS approach. Then, in order to assess which combination of
learning rate—number of iterations of the gradient descent is the best for a regression

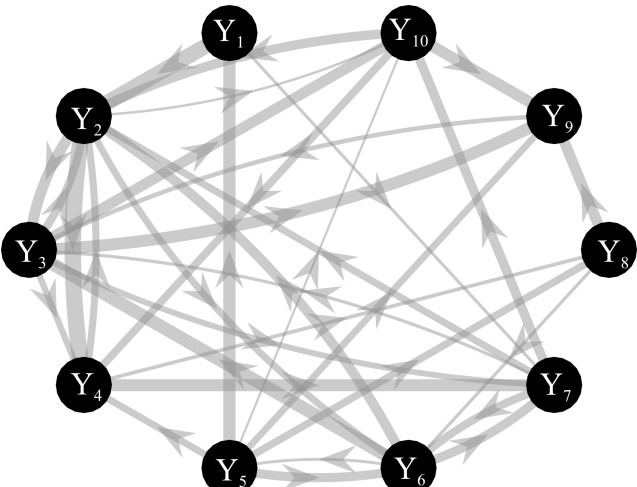

**Figure 3 Graphical representation of one of the ground-truth networks of the simulation study III.** Arrows represent the causal links randomly assigned between two network nodes via nonzero VAR coefficients. The thickness of each arrow is proportional to the strength of the causal connection assessed by the conditional GC, with minimum and maximum values equal to 0.03 and 0.31. The number of connections for each network is set to ~38 out of 90.

problem different measures of performances were computed as explained in the following subsection. Second, by using the best combination of learning rate\ number of iterations of the gradient descent, the effects of $K$ ratio and SNR were assessed by comparing the performances of ANN and OLS in estimating conditional GC. In the latter case, the same multivariate time series generated for the purposes of the first simulation study were used, by simply adding white noise with amplitude tuned to get the desired SNR value.

### Simulation study III

Simulated multivariate time series ($M$=10) were generated as a realization of a reduced VAR(6) process in which coefficients of a VAR(1) model were placed in the first lag for the diagonal elements, while coefficients of a VAR(2) model were placed randomly with a variable delay (up to 6) for the off-diagonal elements (*Rodrigues & Andrade, 2015*). One-hundred surrogate networks were created assuming links in 80% of all possible connections and directed interactions were placed in a subset of existing links (50%), with a final value of density of connected nodes ~40%. Interactions were generated by randomly assigning both positive and negative values to the VAR(2) coefficients outside the diagonal. The magnitude of AR coefficients was randomly determined (range: 0.15–0.5 in steps of 0.01) (*Pascucci, Rubega & Plomp, 2020*). For each simulated dataset, the stochastic generation of a VAR model was reiterated until the system reached the asymptotic stability for which the real eigenvalues are lower than zero (*Barnett & Seth, 2014*). The knowledge of the true AR parameters allows computing the theoretical values of the conditional GC and the true network topology as illustrated in Fig. 3. Simulations were generated for different values of the $K$ ratio, as defined in the previous section, in the range (1, 3, 10, 20) with a resulting time series length $N = 60$ when $K = 1$ and $N = 1200$ when $K = 20$. In order to evaluate the differences between ANN and OLS

estimation approaches, different measures of performance were computed as explained in the following subsection. For the ANN case we used the best combination of learning rate number of iterations of the gradient descent obtained from simulation study I.

### Performance evaluation

Performances were assessed both in terms of the accuracy in estimating the strength of the network links through the absolute values of the conditional GC measure, and in terms of the ability to reconstruct the network structure through the assessment of the statistical significance of the GC.

The bias of GC was computed comparing the estimated and theoretical GC values. For each pair of network nodes represented by the processes $Y_i$ and $Y_j$, the theoretical GC obtained from the true VAR parameters, $F_{i \to j|s}$, was compared with the corresponding estimated GC value, $\widehat{F}_{i \to j|s}$ through the absolute bias measure (*Kim & Kim, 2016*):

$$bias = |F_{i \to j|s} - \widehat{F}_{i \to j|s}|. \tag{19}$$

The bias was assessed separately for null links and non-null-links, corresponding respectively to zero and non-zero values of the conditional GC, yielding the measures $bias_0$ and $bias_1$. *For each* network, these two measures were averaged across the non-null links (15 for the simulations I-II and 38 for the simulation III) and across the null links (75 for the simulations I-II and 52 for the simulation III) to get individual measures, denoted as $BIAS_1$ and $BIAS_0$. Finally, the distributions of the two parameters were obtained across the 100 simulated network structures.

The ability of ANN and OLS to detect the absence or presence of a network link based on the statistical significance of the GC was tested comparing two adjacency matrices representative of the estimated and theoretical network structures. This can be seen as a binary classification task where the existence (class 1) or absence (class 0) of a causal connection is estimated using surrogate data for OLS and looking at the presence/absence of non-zero weights for ANN, and is then compared with the underlying ground-truth structure. Performances were assessed through the computation of false-negative rate (FNR, measuring the fraction of non-null links with non-significant estimated GC), false-positive rate (FPR, measuring the fraction of null links with significant estimated GC) and Area Under Curve (AUC) that summarizes the information provided by FNR and FPR (*Toppi et al., 2016b*; *Antonacci et al., 2019a*). In particular, the AUC parameter is obtained by applying a trapezoidal interpolation between a point on the Receiver Operating Characteristic (ROC) space, extracted knowing false positives and true positives, and the two extremes of the ROC space (0,0) and (1,1). These performance measures were computed across the network links for each assigned network, and the corresponding distribution across the 100 simulated network structures was then obtained separately for OLS and ANN. In the case of ANNs, the computation time (in seconds) required for the training of the ANN for different values of learning rate, number of iterations of the gradient descent and data samples available was also considered as a performance parameter. The average computation times over the 100 realizations were calculated using

an implementation of the algorithms in MATLAB® environment on a PC with a six cores Intel Xeon (CPU clock speed 3.7 GHz), 128·GB DDR4 RAM.

To establish which combination of learning rate and number of iterations of the gradient descent guarantees the most accurate results for each value of the $K$-ratio, an indicator of the overall performance (parameter $S$) was defined as the average of the two following performance parameters: (i) the bias as defined in (19) for non-null links, normalized with respect to the theoretical GC value; (ii) the complement to 1 of the AUC parameter, $1-AUC$. These two parameters are both null in the case of perfect estimation, and increase when the estimated GC values deviate from the theoretical (non-zero) values or when the estimated network topology differs from the true topology. Both parameters were averaged across values of the $K$-ratio, and then the $S$ parameter was computed as their average. The distribution of $S$ across the 100 realizations was investigated as a function of learning rate and number of iterations of SGD-$l_1$.

### Statistical analysis

For the first simulation, a three-way repeated-measures ANOVA was carried out for each performance parameter ($BIAS_0$,$BIAS_1$,$FNR$,$FPR$,$AUC$), in order to evaluate the effects on the computed performance parameters of different values of $K$ (in the range [20, 10, 3, 1]), different values of the learning rate LR (in the range [$10^{-3}$,$10^{-4}$,$10^{-5}$]) and different values of the number of iterations of SGD-$l_1$ ($N_{train}$ in the range [100, 1000, 2000]). Furthermore, with the aim of defining the best combination of learning rate and number of SGD-$l_1$ iterations independently of the data size, a two-way repeated-measures ANOVA was carried out for the parameter $S$ using LR and $N_{train}$ as factors and grouping data from all values of $K$, so as to evaluate the effects of these two parameters on the overall performance.

For the second simulation, five different three-way repeated-measures ANOVA tests, one for each performance parameter ($BIAS_0$,$BIAS_1$,$FNR$,$FPR$,$AUC$), were performed to evaluate the effects on the performance of different values of $K$ (in the range [20, 10, 3]), of different values of $SNR$ (in the range [0.1, 1, 5, 10, $10^3$]) and of the two estimation methods ([OLS, ANN]).

For the last simulation, five different repeated measures two-way ANOVA tests, one for each performance parameter ($BIAS_0$,$BIAS_1$,$FNR$,$FPR$,$AUC$), were performed to evaluate the effects on the performance different values of $K$ (in the range [20, 10, 3]) and different estimation methods ([OLS, ANN]).

The Greenhouse–Geisser correction for the violation of the spherical hypothesis was used in all analyses. The Tukey's posthoc test was used for testing the differences between the sub-levels of the ANOVA factors. The Bonferroni-Holm correction was applied for multiple ANOVAs computed on different performance parameters.

### Results of the simulation study I

The results of the three-way repeated-measures ANOVAs, expressed in terms of F-values and computed separately on all the performance parameters considering K, LR and $N_{train}$ as main factors, are reported in Table 1.

**Table 1 F-values and corresponding degrees of freedom (DoF) of the three-way repeated measures ANOVA.** ***$p < 10^{-5}$; **$10^{-5} < p < 0.01$; *$0.01 < p < 0.05$.

| Factors | DoF | $BIAS_0$ | $BIAS_1$ | FNR | FPR | AUC |
|---|---|---|---|---|---|---|
| $N_{train}$ | (2, 198) | 7.8*** | 711*** | 467*** | 68*** | 609*** |
| LR | (2, 198) | 69.6*** | 461*** | 325*** | 171*** | 656*** |
| K | (3, 297) | 16*** | 181*** | 309*** | 88*** | 344*** |
| $N_{train} \times LR$ | (4, 396) | 110.4*** | 101*** | 279*** | 156*** | 97.2*** |
| $N_{train} \times K$ | (6, 594) | 139.7*** | 2.6* | 44*** | 98*** | 0.5 |
| $LR \times K$ | (6, 594) | 200.9*** | 13*** | 47*** | 132*** | 2.5* |
| $N_{train} \times LR \times K$ | (12, 1,188) | 28.2*** | 71.6*** | 20*** | 15*** | 3.6*** |

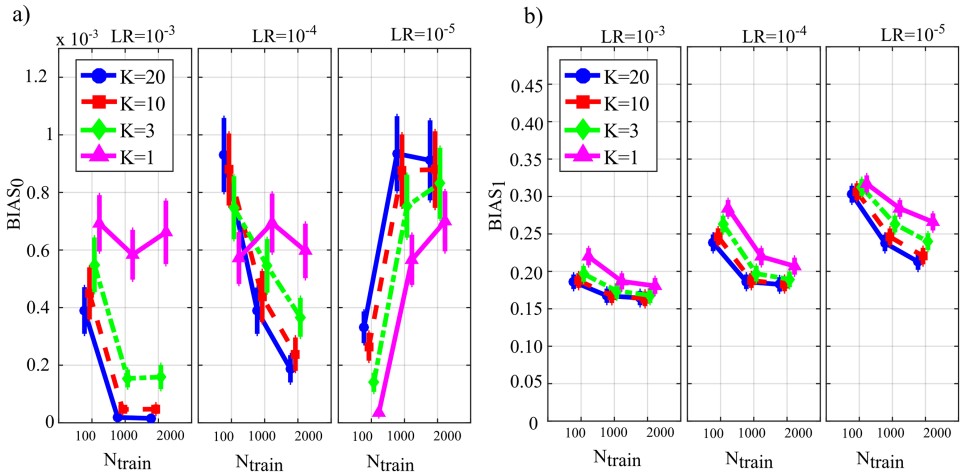

**Figure 4 Distributions of the bias of conditional GC (value and 95% confidence interval across 100 simulated networks) estimated using ANNs for the first simulation study.** Bias parameters computed for the null links (BIAS0), (A) and for the non-null links (BIAS1), (B) are plotted as a function of the number of iterations of the gradient descent (Ntrain) for different values of the ratio between data samples and model coefficients to be estimated (K) and of the learning rate (LR) of ANN training.

The three-way ANOVAs revealed a strong statistical influence of the main factors $N_{train}$, $LR$ and $K$ and of their interaction on all the performance parameters analyzed. The only non-significant effect was that of the interaction between $N_{train}$ and $K$ on the AUC parameter.

Figure 4 reports the distribution of the parameters $BIAS_1$ and $BIAS_1$ according to the interaction $N_{train} \times LR \times K$. In the analysis of the error associated with the estimation of the conditional GC along the null links ($BIAS_0$, Fig. 4A), an increase of the bias was observed at decreasing the number of data samples available (factor $K$), regardless of the learning rate (factor $LR$) and of the number of iterations of gradient descent ($N_{train}$).

Except for the case $LR = 10^{-5}$, increasing the number of iterations $N_{train}$ reduced the bias for $LR = 10^{-3}$ and for $LR = 10^{-4}$, but not for $LR = 10^{-5}$ when the opposite behavior was observed. The bias analysis of the GC values computed along the non-null links (Fig. 4B) showed more clear patterns of the error, evidencing a decrease of $BIAS_1$ at

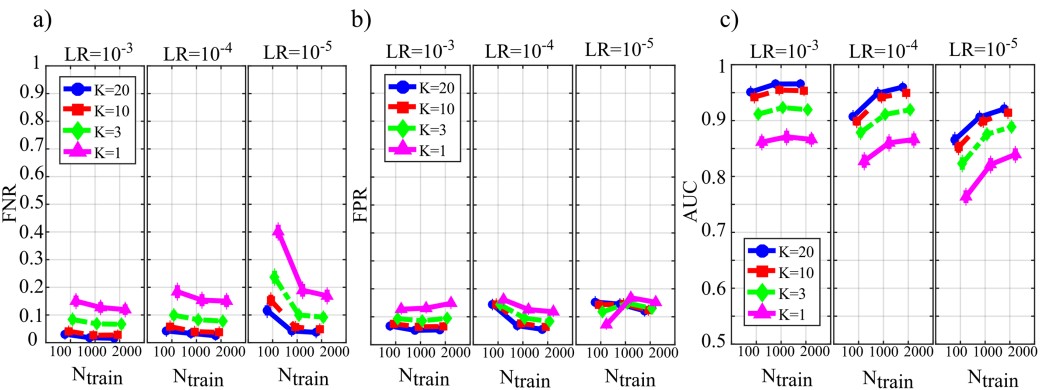

**Figure 5 Distributions of the parameters assessing the quality of network reconstruction performed using ANNs for the first simulation study.** Plots depict the distributions of FNR (A), FPR (B) and AUC (C) expressed as mean value and 95% confidence interval across 100 simulated networks as a function of the number of iterations of the gradient descent (Ntrain) for different values of the ratio between data samples and model coefficients to be estimated (K) and of the learning rate (LR) of ANN training.

increasing $N_{train}$, at increasing $K$, and at decreasing $LR$. The lowest mean values of $BIAS_1$ were obtained setting $LR = 10^{-3}$ and $N_{train}$ equal to 1,000 or 2,000.

Figure 5 reports the distributions of the parameters FNR, FPR and AUC according to the interaction $N_{train} \times LR \times K$. The portion of non-null directed links incorrectly classified as null (*FNR*, Fig. 5A) was lower than 20% in all cases except for $N_{train} = 100$ and $K \leq 3$. The rate of false negative detections decreased at increasing $K$ regardless of $LR$ and $N_{train}$. A strong effect of the number of iterations on the *FNR* was observed in the most challenging condition of $K = 1$ (purple lines), especially when $LR = 10^{-5}$. The portion of null links incorrectly classified as non-null (*FPR*, Fig. 5B) was always lower than 20%. The rate of false positive detections showed a tendency to increase at decreasing $K$, while it was almost stable at varying $LR$ and $N_{train}$. The best scenario appears $LR = 10^{-3}$, showing a mean *FPR* under 0.1 for each value of $K > 1$. The overall accuracy measured by AUC (Fig. 5C) reached the highest values for $LR = 10^{-3}$ and $N_{train} \in \{1,000, 2,000\}$. In these conditions, a very accurate reconstruction of the network structure was obtained, as the accuracy was equal to 95 % for $K = 20$ and above 85% even when $K = 1$. The performance showed a tendency to degrade at decreasing $K$, increasing $LR$ and decreasing $N_{train}$.

Table 2 reports the computation time required for the training of the neural network in different conditions of $K$ ratio, learning rate and number of SGD-$l_1$ iterations averaged across the 100 realizations. As expected, the computation time increases with the number of iterations of the gradient descent and with the number of data samples available ($K$ ratio). The least and most time-consuming settings were $N_{train} = 100$, $K = 1$ and $N_{train} = 2,000$, $K = 20$, respectively taking ~2 s and ~210 s.

Figure 6 reports the distribution of the overall performance parameter $S$ computed as a function of the learning rate for different number of iterations of SDG-$l_1$ (interaction $N_{train} \times LR$). The results show how the performance is affected significantly by both factors, with values of $S$ that tend to decrease while increasing the learning rate and the number of

**Table 2 Average computation time (in seconds, measured for 100 simulated networks) required to train the ANN for different values of K ratio, learning rate and number of iteration of gradient descent.**

| $N_{train}$ | $LR = 10^{-3}$ | | | $LR = 10^{-4}$ | | | $LR = 10^{-5}$ | | |
|---|---|---|---|---|---|---|---|---|---|
| | 100 | 1,000 | 2,000 | 100 | 1,000 | 2,000 | 100 | 1,000 | 2,000 |
| K = 20 | 12.08 | 107.7 | 213.66 | 12 | 107.7 | 214.36 | 11.91 | 107.8 | 213.72 |
| K = 10 | 7.6 | 72.8 | 145.1 | 7.68 | 72.8 | 145.1 | 7.61 | 72.88 | 145.28 |
| K = 3 | 3.4 | 33.12 | 65.9 | 3.44 | 33.25 | 66.1 | 3.4 | 33.18 | 66.22 |
| K = 1 | 2.6 | 25.9 | 51.7 | 2.64 | 25.98 | 51.69 | 2.6 | 26 | 51.82 |

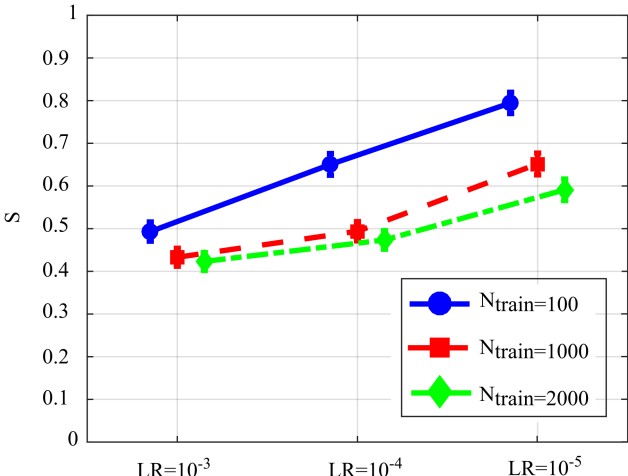

**Figure 6 Distributions of S parameter considering the interaction factor $N_{train} \times LR$, expressed as mean value and 95% confidence interval of the parameter computed across 100 realizations of the first simulation study ($F(4,396) = 128.09$, $p < 10^{-5}$).**

iterations of the gradient descent. The lower values of $S$, indicating lowest bias of the estimated GC values and/or highest AUC in the classification of the network structure, were observed for $LR = 10^{-3}$ and $N_{train} = 1,000$ or $N_{train} = 2,000$. As the improvement from $N_{train} = 1,000$ to $N_{train} = 2,000$ was not statistically significant, we infer that the best setting is the least computationally onerous combination, i.e., $LR = 10^{-3}$, $N_{train} = 1,000$.

### Results of the simulation study II

After the extraction of the best combination of the training parameters of the ANN, in the second simulation study we compare the performance of OLS and ANN at varying the proportion between number of data samples available and parameters to be estimated (K-ratio) as well as at varying the amplitude of white noise added to the original time series (SNR). The results of the three-way repeated-measures ANOVAs, expressed in terms of F-values and computed separately on all the performance parameters considering $K$, $SNR$ and $TYPE$ (i.e., the method used: OLS or ANN) as main factors, are reported in Table 3.

**Table 3 F-values and corresponding degrees of freedom (DoF) of the three-way repeated measures ANOVA investigating the effects of the factors K (ratio between data samples and number of model parameters), SNR (ratio between the squared amplitude of the signal and the square amplitude of the noise) and TYPE (estimator used, i.e., OLS or ANN) on the performance parameters of GC estimation (BIAS0, BIAS1) and of network reconstruction (FNR, FPR, AUC). \*\*\*$p < 10^{-5}$; \*\* $10^{-5} < p < 0.01$.**

| Factors | DoF | $BIAS_0$ | $BIAS_1$ | FNR | FPR | AUC |
|---|---|---|---|---|---|---|
| **TYPE** | (1, 99) | 5,901*** | 77.8*** | 68.8*** | 27.4*** | 36.9*** |
| **SNR** | (4, 396) | 328.1*** | 1,621.4*** | 645.1*** | 173.3*** | 761.2*** |
| **K** | (2, 198) | 9,785.3*** | 0.2 | 2,118.7*** | 10.2*** | 1881.1*** |
| **TYPE × SNR** | (4, 396) | 85.6*** | 199.7*** | 2.6** | 46.2*** | 10.5*** |
| **TYPE × K** | (2, 198) | 8,578*** | 99.9*** | 1,093*** | 280.8*** | 570.1*** |
| **SNR × K** | (8, 792) | 33.3*** | 167.4*** | 50.4*** | 19.4*** | 30.3*** |
| **TYPE × K × SNR** | (8, 792) | 26.1*** | 128.8*** | 65.4*** | 13.3*** | 45.4*** |

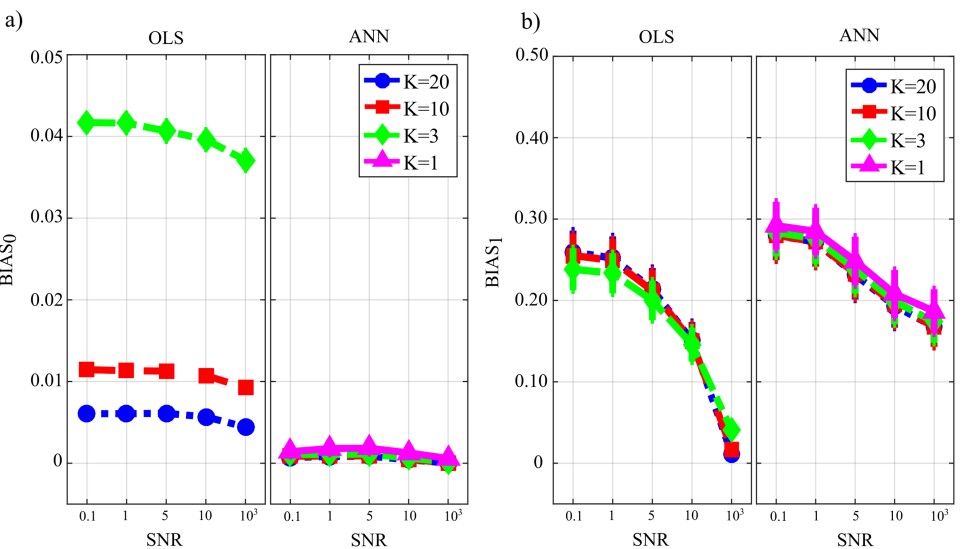

**Figure 7 Distributions of the bias relevant to the estimation of GC on the null links ($BIAS_0$), (A) and on the non-null links (BIAS1), (B) plotted as a function of the ratio between data samples available and number of parameters to be estimated (K) and of the ratio between signal amplitude and noise amplitude (SNR), for OLS estimation and ANN estimation.**

The three-way ANOVA highlights a strong statistical influence of the main factors *K*, *SNR* and *TYPE* and of their interactions on all the performance parameters analyzed in this study. In this case the level $K = 1$ was not considered in the statistical comparison due to the non-convergence of the DARE equation for the OLS case.

Figure 7 reports the distribution of the parameters $BIAS_0$ and $BIAS_1$ according to the interaction factor $TYPE \times K \times SNR$. The comparison of OLS and ANN shows that the two estimation approaches have very different performance: in the computation of GC over the null links, the error of ANN is very close to zero even in the most challenging condition of $K = 1$, while OLS shows an increasing bias with the decrease of the number of

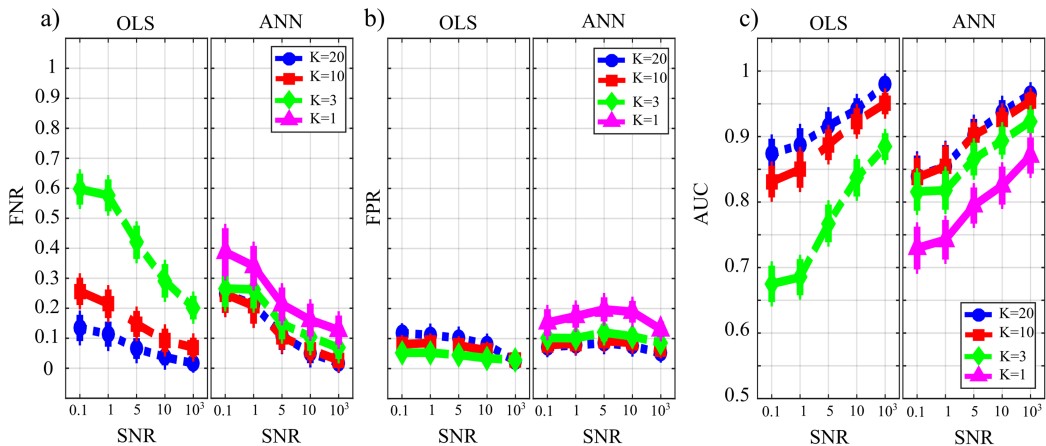

**Figure 8 Distributions of the parameters assessing the performance of network reconstruction, i.e. the rate of false negatives (FNR), (A) and of false positives (FPR), (B) and of the area under the curve (AUC), (C) plotted as a function of the ratio between data samples available and number of parameters to be estimated (K) and of the ratio between signal amplitude and noise amplitude (SNR), for OLS estimation and ANN estimation.**

data samples available for the estimation of GC values (Fig. 7A); in the computation of GC over the non-null links, the estimation bias is low but shows a tendency to increase for OLS, while it is remarkable but stable for the ANN. Concerning the additive noise, its impact is much more noticeable for the OLS case which shows a large increase of the bias measures with the decrease of SNR values; on the other hand, the trends of the two measures of bias for the ANN case seem to be rather constant. Only the bias in the computation of the GC on non-null links shows a slight reduction with increasing SNR. However, in a condition of sufficient data samples available (K = 20) and a high value of signal-to-noise ratio ($SNR = 10^3$), OLS shows a bias associated with the non-null links which is very close to zero and considerably lower than that associated with ANN.

Figure 8 reports the distributions of the parameters FNR, FPR and AUC according to the interaction $TYPE \times K \times SNR$. When the value of SNR is equal to $10^3$ the analysis of false negative detections of directed links (panel a) shows that the error committed increased with decreasing the number of data samples available. The error was comparable for OLS and ANN when $K = [20,10]$, and then increased more markedly for OLS, while it remained lower than 10% even when $K = 1$ for ANN. On the other hand, the analysis of false positive detections (panel b) showed an error quite low and stable with $K$ in the case of OLS, and an error slightly growing with $K$ up to 15% in the case of ANN. The overall performance evaluated through AUC showed high classification accuracy and absence of statistically significant differences between the two estimation methods for $K = [20,10]$, and a better performance of ANN compared with OLS for lower values of $K$; a high AUC value (~85%) was reported for ANN even when K = 1. The situation becomes very different when the value of SNR decreases. Both false negatives and false positives increase with the amplitude of the additive noise; the increase of FNR is remarkable for the OLS method. The analysis of AUC trends for OLS case (panel c) highlights that when SNR is very low and the number of data samples available is very scarce (K = 3, green line)

**Table 4 Average computation time (in seconds, measured for 100 simulated networks) required by the OLS and ANN methods for the estimation of GC at different values of K ratio (SNR = 103).**

| Method | OLS | ANN |
|---|---|---|
| $K = 20$ | $9.1 \cdot 10^3$ | 142.18 |
| $K = 10$ | $4.5 \cdot 10^3$ | 107.28 |
| $K = 3$ | $1.3 \cdot 10^3$ | 67.6 |
| $K = 1$ | – | 60.38 |

**Table 5 F-values and corresponding degrees of freedom (DoF) of the two-way repeated measures ANOVA investigating the effects of the factors K (ratio between data samples and number of model parameters) and TYPE (estimator used, i.e., OLS or ANN) on the performance parameters of GC estimation (BIAS0, BIAS1) and of network reconstruction (FNR, FPR, AUC). ***$p < 10^{-5}$; ** $10^{-5} < p < 0.01$.**

| Factors | DoF | $BIAS_0$ | $BIAS_1$ | FNR | FPR | AUC |
|---|---|---|---|---|---|---|
| **TYPE** | (1, 99) | 1,295*** | 3,518*** | 105.7*** | 491*** | 174.4*** |
| **K** | (2, 198) | 7,454*** | 1,196.2*** | 968.2*** | 111.1*** | 1,468*** |
| **TYPE × K** | (2, 198) | 6,770.5*** | 15.8** | 268.5*** | 69.5*** | 102.8*** |

AUC is less than 70 %. This is not the case for ANN which shows an average value of AUC greater than 70 % even when K = 1 (purple line) and SNR = 0.1 which represents the worst simulated scenario. Using a quantile-based thresholding criteria approach for the AUC computation, as introduced in (*Pascucci, Rubega & Plomp, 2020*), yields substantially overlapping trends of the performance measures (results reported in Fig. S1 as Supplementary Material).

Table 4 reports the computation time required for the entire process of GC computation using the two estimation approaches for different values of the $K$ ratio when $SNR = 10^3$. OLS analysis includes SS model identification and the subsequent evaluation of the null-case distribution for each couple of nodes as described in the Methods section. ANN analysis includes SS model identification plus the training process at $N_{train} = 1,000$, $LR = 10^{-3}$. The analysis highlights the expected decrease of the computation times with decreasing the $K$ ratio and, more importantly, a strong reduction of the time requested for the entire process when ANN is used in place of OLS. The computation time of OLS identification is not reported for $K = 1$ due to the non-convergence of the solution to the DARE equation necessary for SS model identification.

### Results of the simulation study III

In the last simulation study, we compare the performance of OLS and ANN at varying the proportion between the number of data samples available and parameters to be estimated ($K$-ratio). The results of the two-way repeated-measures ANOVAs, expressed in terms of F-values and computed separately on all the performance parameters considering $K$ and *TYPE* (i.e., the method used: OLS or ANN) as main factors, are reported in Table 5. The two-way ANOVA analysis highlights a strong statistical influence of the

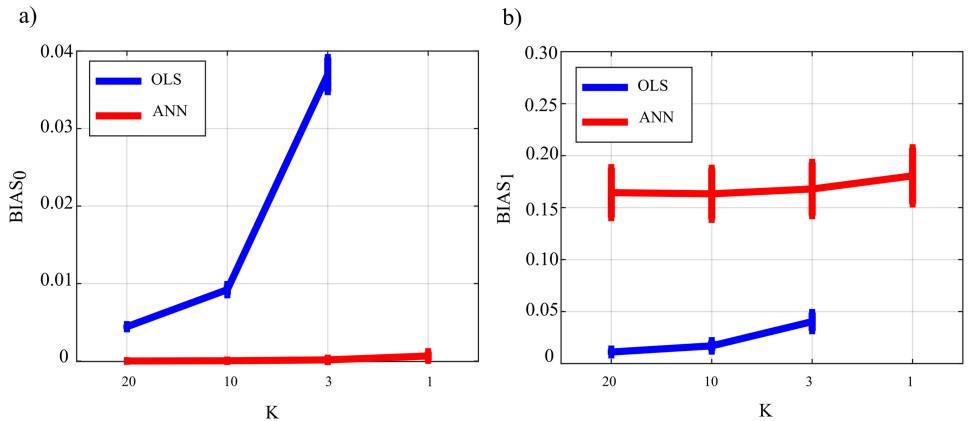

**Figure 9 Distributions of the bias relevant to the estimation of GC on the null links ($BIAS_0$), (A) and on the non-null links ($BIAS_1$), (B) plotted as a function of the ratio between data samples available and number of parameters to be estimated ($K$), for OLS estimation (blue) and ANN estimation (red).**

main factor K and TYPE and of their interaction ($TYPE \times K$) on all the performance parameters analyzed. Also in this case the level $K = 1$ was not considered in the statistical analysis due to the non-convergence of the DARE equation for the OLS case.

Figure 9 reports the distribution of the parameters $BIAS_0$ and $BIAS_1$ according to the interaction factor $K \times TYPE$. The comparison of OLS (blue line) and ANN (red line) confirms the trends obtained for the case $SNR = 10^3$ in the Simulation study II. In fact, the bias associated with ANN in the computation over null links is very close to zero even in the most challenging condition of $K = 1$ with OLS showing a very different trend with a strong increase associated with decreasing K-ratio values (panel a); in the computation over the non-null links for ANN, the estimation bias displays a tendency to be stable but remarkable if compared with OLS case.

Figure 10 reports the distribution of FNR, FPR, and AUC according to the interaction $K \times TYPE$. The analysis of both false negatives (panel a) and false positives (panel b) shows a decrease with the increase of the number of data samples available. The false negative rate is comparable for OLS and ANN when $K = [20,10]$, and then increases particularly for OLS while for ANN it assumes an average value around 30% in the most challenging situation ($K = 1$). The analysis of false positives (panel b) shows a quite low and stable trend for the OLS case for all values of $K$, and an increasing trend for ANN up to 20% ($K = 1$). Even in the best scenario of $K = 20$ the false positive rate assumes an average value of ~10%. The overall performance evaluated through AUC indicates high classification accuracy ($\geq$95%) with a statistically significant difference between the two methods when $K = [20,10]$ and a comparable performance when $K = 3$ with no statistically significant differences highlighted by the post-hoc test. However, in the most challenging situation of $K = 1$ the ANN method leads to an AUC value greater than 75 %. As a general remark, there is a worsening of the performance in reconstructing the GC network if compared with a sparser simulated network (Study II) that is more evident in the ANN case.

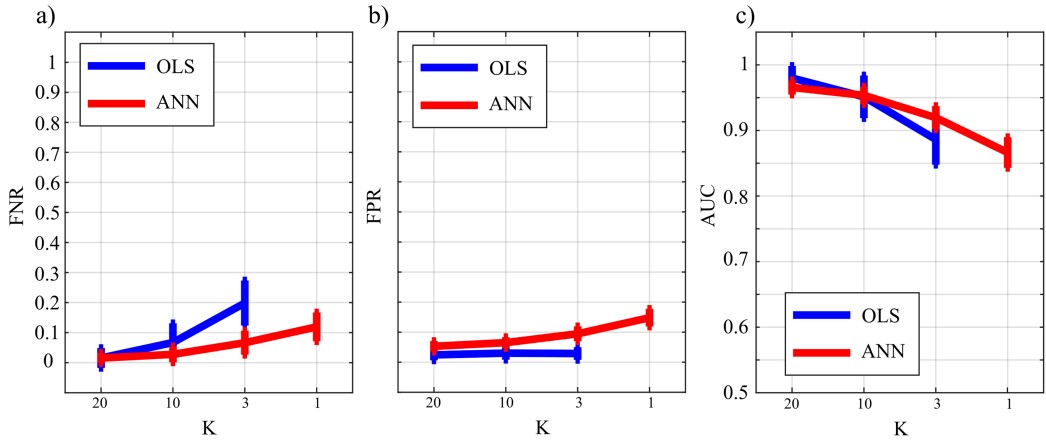

**Figure 10 Distributions of the parameters assessing the performance of network reconstruction, i.e., the rate of false negatives (FNR), (A) and of false positives (FPR), (B) and of the area under the curve (AUC), (C) plotted as a function of the ratio between data samples available and number of parameters to be estimated ($K$) for OLS estimation (blue) and ANN estimation (red).**

# APPLICATION TO PHYSIOLOGICAL TIME SERIES

This section reports the application of the conditional GC, defined as in Eq. (12) and computed using OLS and ANN estimators, to the analysis of physiological networks formed by several time series reflecting the variability of heart rate, respiration, blood pulse propagation time, and of the amplitudes of different brain waves detected from EEG signals. The dataset used for the analysis was collected in a previous study on the interactions between various organ systems during different levels of mental stress (*Zanetti et al., 2019*).

## Data acquisition and pre-processing

The experimental protocol involved eighteen healthy participants with age between 20 and 30 years, from whom different physiological signals were recorded during three tasks inducing different levels of mental stress: a resting condition lasting 12 min and consisting in watching a relaxing video (R); a mental arithmetic test during which the volunteer had to carry out the maximum number of 3-digit sums and subtractions (M); a sustained attention task that consisted in following a cursor on the screen while trying to avoid some obstacles (G). The experiment was approved by the Ethics Committee of the University of Trento, and all participants provided written informed consent. The study was in accordance with the Declaration of Helsinki.

The acquired physiological signals were the Electrocardiogram (ECG) signal, the respiratory signal (RESP) monitoring abdomen compartment movements, the blood volume pulse (BVP) signal measured through a photoplethysmographic technique, and Electroencephalogram (EEG) signals acquired using 14 channels Emotiv EPOC PLUS (international 10–20 locations). More details on the instrumentation and acquisition steps can be found in (*Zanetti et al., 2019*). The acquired physiological signals, representing the dynamical activity of different integrated physiological systems, were processed to

extract synchronous time-series representing the time-course of different stochastic processes. Specifically, a template matching algorithm was employed to extract R peaks from the ECG and then measure R-R interval time series (process $\eta$). The breath signal was sampled in correspondence of the R peaks to attain respiratory time series (process $\rho$). Moreover, the pulse arrival time was extracted as the time interval between the ECG R peak and the maximum derivative of the BVP signal (process $\pi$) for each cardiac cycle. With regard to brain activity, the power spectral density (PSD) of the EEG signals measured at the electrode $F_z$ was calculated using a 2-s long sliding window with 50% overlap. Then, for each window, the PSD was integrated within four different frequency bands to obtain time series representative of the $\delta$ (0.5–3 Hz), $\theta$ (3–8 Hz), $\alpha$ (8–12 Hz) and $\beta$ (12–25 Hz) brain wave amplitudes. The use of these frequency bands was motivated by studies which relate increasing levels of fatigue or alertness with higher PSD of the $\delta$, $\theta$ and $\alpha$ processes and lower PSD of the $\beta$ process (*Sciaraffa et al., 2020*; *Tran et al., 2007*; *Trejo et al., 2007*). The brain time series extracted in this way was synchronous with those obtained resampling at 1 Hz the three cardiovascular time series using spline interpolation (*Zanetti et al., 2019*). The rate of 1 Hz, which sets a time scale for the analysis which is compatible with the spectrum of heart rhythms, has already been used in previous studies in the field of network physiology for analyzing the time series from different body locations (*Bashan et al., 2012*; *Bartsch et al., 2015*). The uniformity of the final sampling rate and the synchronization of the signals acquired from different devices permitted to obtain seven synchronous time series for all the physiological districts.

Following the procedure described above, synchronous segments of the seven time series were selected inside each experimental condition (R, M, or G); each time series consisted of 300 samples, corresponding to five minutes of signal recording. All time series were checked for a restricted form of weak sense stationarity using the algorithm proposed in (*Magagnin et al., 2011*), which randomly extracts a given number of sub-windows from each time series and assesses the steadiness of mean and variance across the sub-windows. The seven time series extracted from each subject and experimental condition were considered to be a realization of a VAR process descriptive of the behavior of a dynamical system that describing the observed network of physiological interactions. For each subject and condition, the parameters of the VAR model fitting the seven observed time series, $A_1,\ldots, A_p, \sigma$, were estimated with the two procedures described (i.e., OLS and ANN). The model order $p$ was estimated for each experimental condition and subject through the Bayesian Information Criterion (BIC) (*Schwarz, 1978*).

### Granger Causality analysis

To assess the topological structure of the physiological network, the conditional Granger causality between each pair of nodes, $F_{i\rightarrow j|s}$, was computed through SS analysis applied to the VAR parameters estimated with the two presented methods (i.e., OLS and ANNs), and its statistical significance was assessed with the associated approach (i.e., using surrogate data for OLS and exploiting the intrinsic sparseness after the training process for ANN). The analysis was performed between each pair of processes as driver and target ($i,j = [\eta,\rho,\pi,\delta,\theta,\alpha,\beta]$, $i \neq j$) and collecting the remaining five processes in the

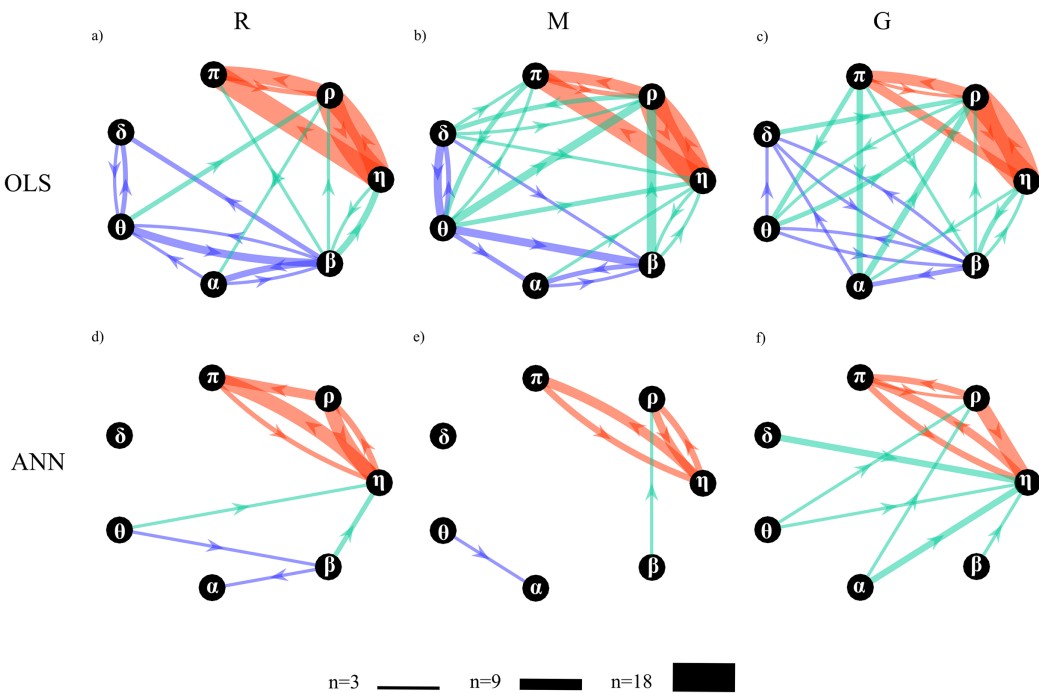

**Figure 11 Topological structure of the network of physiological interactions reconstructed during the rest (R), mental arithmetic (M) and serious game (G) experimental conditions.** Graphs depict significant directed interactions within the brain (purple arrows), body (red arrows) and brain-body (green arrows) sub-networks. Directed interactions were assessed counting the number of subjects for which the conditional Granger causality ($F_{i \to j|s}$) was detected as statistically significant using OLS (A–C) or ANN (D–F) in the estimation process. The arrow thickness is proportional to the number of subjects ($n$) for which the link is detected as statistically significant.

conditioning vector with index $s$. Moreover, to confirm the results obtained in (*Antonacci et al., 2020b*) on the same data, the in-strength—defined as the sum of all weighted inward links (*Rubinov & Sporns, 2010*)—was computed for a specific network node (pulse arrival time $\pi$). The effect of the different experimental conditions on the in-strength evaluated for the $\pi$ node was assessed through the Kruskal-Wallis test followed by the Wilcoxon rank-sum test between pairs of conditions. All analyses were performed with a model of dimension $Mp$, where $M = 7$ and $p \sim 4$ (depending on the BIC) on time series of 300 points, corresponding to $K \sim 10$ relating the amount of data sample available to the model dimension. The performed analysis can be replicated by running the MATLAB script Test_Application in the released toolbox for a single subject taken from the entire dataset (TimeSeriesStress).

## Results of Granger causality analysis

Figure 11 depicts the network of physiological interactions reconstructed through the detection of the statistically significant values of the conditional Granger causality ($F_{i \to j|s}$) computed for all pairs of processes belonging to the analyzed network. The weighted arrows represent the most active connections among the systems (arrows are present when at least three subjects show a statistically significant value of $F_{i \to j|s}$). To ease interpretation

and comparison between OLS and ANN estimates, the three sub-networks representative of brain, body and brain-body interactions are depicted with arrows of different colors. The networks estimated using OLS in the three experimental conditions (Figs. 11A–11C) exhibit similar structures to those estimated using ANN (Figs. 11D–11F); the main difference is that networks estimated with ANN show greater sparsity than those estimated with OLS.

A qualitative analysis of the networks illustrates the existence of a highly connected body sub-network (red arrows), a weakly connected brain sub-network (purple arrows), and a pattern of brain-body interactions (green arrows) that changes with the experimental condition. The body interactions are characterized, consistently across the three conditions, by cardiovascular links (interactions from $\eta$ to $\pi$) and cardio-respiratory links (interactions between $\eta$ and $\rho$), with a weaker coupling between $\rho$ and $\pi$. The use of ANN reveals a preferential direction from $\rho$ to $\pi$ that is not present in the condition M and is bidirectional in the condition G. The topology of the brain sub-network assessed by the ANN method is less stable across conditions, and looses consistency moving from R to G. On the contrary, in the OLS case, the topology seems to be more consistent exhibiting weaker connections moving from R to M and from M to G. The analysis of brain-body interactions reveals that such interactions are mostly directed from the brain to the body sub-networks; in this case, the use of ANN clearly shows an increasing of brain-body interactions during the condition G.

Figure 12 reports the distribution of the values of the in-strength index evaluated for the $\pi$ node in each experimental condition. For both OLS and ANN, the median value of the in-strength index is significantly higher in the condition R with respect to the condition G. The use of ANN highlights lower values for the in-strength parameter even if the trend is the same moving across the three experimental conditions. These results show that both approaches detect a decrease of the information flow directed to the cardiovascular node of the body subnetwork, documented by the reduction of the in-strength index in the G condition for the process $\pi$.

## APPLICATION TO A RING OF NON-LINEAR ELECTRONIC OSCILLATORS

In this section we investigate the application of GC, in its unconditional version, computed through OLS and ANN by exploiting the SS approach, to a dataset of electronic non-linear chaotic oscillators, recorded from a unidirectionally-coupled ring of 32 dynamic units, previously realized with the aim of studying remote synchronization (*Minati, 2015a*; *Minati et al., 2018*). In the literature, it has been pointed out that a single transistor oscillator can exhibit very complex activity and a ring of coupled oscillators can create a community structure with statisical properties resembling physiological systems (*Takahashi, 2013*; *Stam, 2005*; *Minati et al., 2015*). The previous analysis has shown how it is possible to provide a mesoscopic description of the information exchanged between different nodes of a network which represents the activity of several physiological systems. On the other hand, the employment of an electronic circuit comprising a ring of oscillators, provides a system of reduced scale and complexity, with respect to a

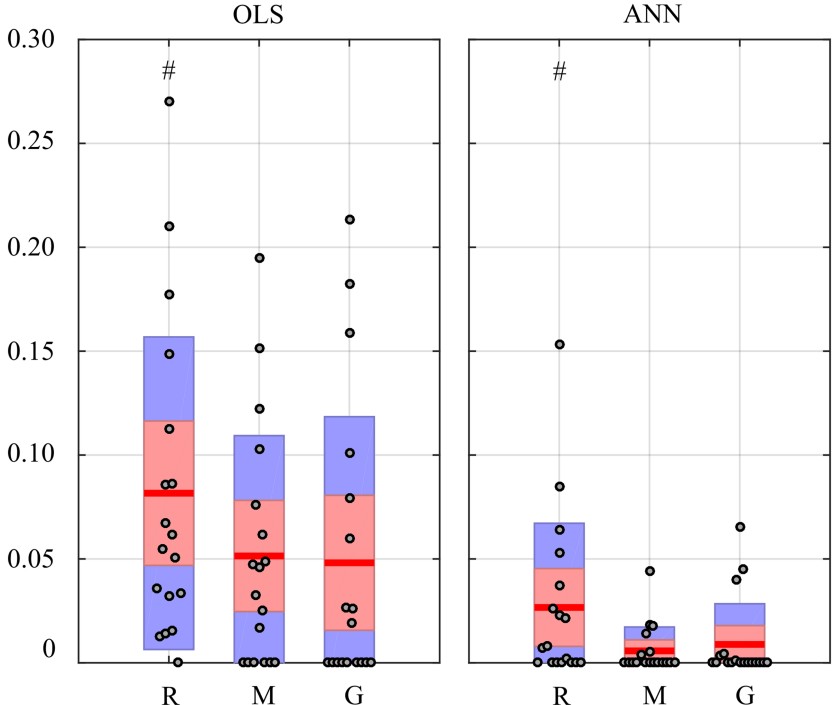

**Figure 12** **In-strength index computed for π node of the physiological network.** Box plots report the distribution across subjects (median: red lines; interquartile range: box; $10^{th}$–$90^{th}$ percentiles: blue bars) and the individual values (circles) of the in-strength computed at rest (R), during mental stress (M) and during serious game (G). Statistically significant differences between pairs of distributions are marked with # (R vs G).

physiological one, yielding full access to the activity of each individual node. The resulting time series, measured as voltage output by each oscillator, were considered as input for a VAR model and for an ANN, descriptive of the behavior of the entire network ring.

## System description and synchronization analysis

The structural diagram of the oscillator circuit corresponding to each node in the network is reported in Fig. 13A and comprises four summing stages associated with low-pass filters. Three such stages with negative gains $G_1 = -3.6$, $G_2 = -3.12$, $G_4 = -3.08$ and filter frequency $F_1 = F_2 = F_3 = 2$ kHz are arranged as a ring oscillator. Two Integrator stages with integration constants $K_1 = 3.67$ $K_2 = 0.11$ $\mu s^{-1}$ with mixing gains $G_3 = -0.5$ and $G_5 = -0.71$ are overlapped to this structure. The ring is completed through fourth summing stages having $F_4 = 100$ kHz :amp:gg; $F_1$ with one input (gain $G_6 = 0.132$) which is necessary to close the internal ring itself and another (gain $G_i = -1.44$) connected to the previous oscillator in the ring network (Fig. 13B). To limit the voltage swing for the off-chip signal a gain inverter $G_0 = -0.4$ is installed. The recorded time series have a length $l = 65,536$ points and are sampled with a sampling frequency $f_s = 100$ kHz and are freely available (*Minati, 2015b*).

The frequency spectrum of each node is represented by three peaks: the most prominent (central one) at $f_c \approx 2.8$ kHz and two weaker ones (sidebands) at $f_l = f_c/2 \approx 1.4$ kHz and

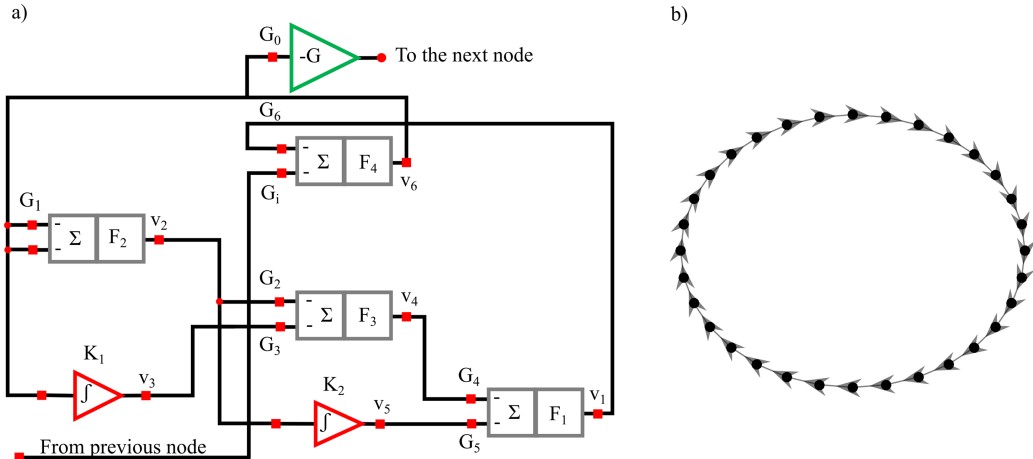

**Figure 13 Diagram of the oscillator circuit corresponding to each node in the network (A). Master-Slave (unidirectional, clock-wise) structure of the ring comprising thirty-two oscillators (B).**

$f_h = f_l + f_c \approx 4.2$ kHz. The higher sideband represents the mirror frequency of the lower one. As explained in (*Minati et al., 2018*), demodulation via envelope detection and subsequent interference occurs, and these phenomena lead to spatial fluctuations of the lower sideband amplitude that are closely related to the remote synchronization effect. In this system, remote synchronization is manifest as a non-monotonic decay of synchronization along the ring, wherein, with increasing distance from a given node, on average synchronization drops, then increases transitorily, and finally vanishes.

As in previous works (*Minati, 2015a*; *Minati et al., 2018*), we determined the instantaneous phase $\phi_m(t)$ and the envelope $A_m(t)$ of the output signal $v_m(t)$ of each oscillator $m$ with the following relationship:

$$v_m(t) + i\hat{v}_m(t) = A_m(t)e^{i\phi(t)}, \tag{20}$$

where $\hat{v}_m(t)$ is the Hilbert transform of the recorded signal $v_m(t)$.

Given two generic time series $Y_i$ and $Y_j$, amplitude synchronization for the envelope $A_m(t)$ was considered in terms of the maximum normalized cross-correlation coefficient for non-negative lags (that is, lags that take into account a possible propagation time along the direction of coupling, clock-wise in this system) $max[C_{ij(\tau)}]_{\tau \geq 0}$ which is defined as:

$$C_{ij}(\tau) = \frac{k_{ij}(\tau)}{\sqrt{\sigma_i^2 \sigma_j^2}}, \tag{21}$$

where $k_{ij}(\tau) = E\left[(Y_{i,n+\tau} - \mu_i)(Y_{j,n+\tau} - \mu_j)\right]$ is the time cross-covariance, $\mu_i = E\left[Y_{i,n}\right]$ and $\mu_j = E\left[Y_{j,n}\right]$ that represent the mean of values of $Y_i$ and $Y_j$; $\sigma_i^2 = \mathbb{E}[(Y_{i,n} - \mu_i)^2]$ and $\sigma_j^2 = \mathbb{E}[(Y_{j,n} - \mu_j)^2]$ which correspond to the variances of $Y_i$ and $Y_j$ respectively.

In Fig. 14 the analysis of cross-correlation coefficient performed for each pair of oscillators $(i,j)$ in the entire ring (panel a) is reported, alongside with the corresponding

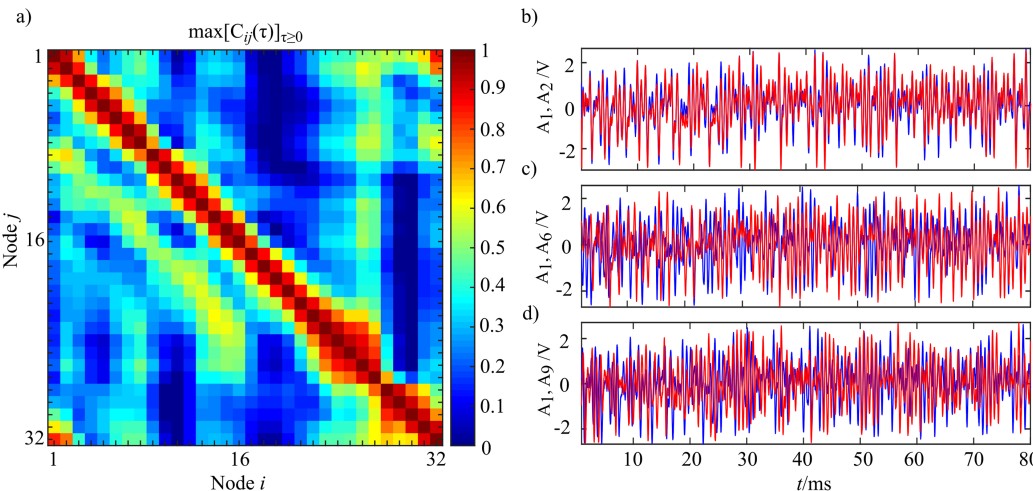

**Figure 14 Instance of remote synchronization.** (A) Reports the synchronization matrix for the entire ring intended as the maximum positive cross-correlation coefficient for the signal envelope $A_m(t)$. (B) Shows the signal envelope $A_m$ for three different coupled of nodes demonstrating remote synchronization effects. The blue line represents $A_1$ with the red line that shows $A_2$ (B), $A_6$ (C) and $A_9$ (D). Time series were realigned to the lag for which the maximum value of cross correlation was observed.

synchronization analysis for three representative oscillator pairs (panel b) which exemplify the decay and transient recovery of amplitude synchronization for three different distances from the node 1. The analysis of the cross-correlation coefficient reveals that moving away from a node, synchronization initial decayed, then gradually increased, rising till a distance $d \approx 8$, and eventually vanished as shown in Fig. 14A. The structural coupling on the ring is only between first neighbors, as indicated by the master-slave configuration, and the highlighted non-monotonic trend in the cross-correlation coefficient indicates a situation of remote synchronization. The visual inspection of signal envelope for three different couples of oscillators (panel b) confirms the analysis of cross-correlation with complete synchronization of the couple $i = 1$, $j = 2$ (distance 1, $max[C_{ij(\tau)}]_{\tau \geq 0} = 0.91$) that becomes a desynchronization for the couple $i = 1$, $j = 6$ (distance 5, $max[C_{ij(\tau)}]_{\tau \geq 0} = 0.19$); finally, the synchronization appears to be strong even for the couple $i = 1$, $j = 9$ that means a physical distance of eight ($max[C_{ij(\tau)}]_{\tau \geq 0} = 0.59$). The performed analysis can be replicated by running the Matlab script Test_Oscillators in the released toolbox.

## Granger Causality analysis

From a theoretical point of view cross-correlation coefficient is a symmetric measure and thus, its value for each time step is the same independently of the selected direction ($i \rightarrow j$, $j \rightarrow i$). For this reason, it is not possible to assess if there is an information exchange between different oscillators. In order to test if there is information exchange between different oscillators, and if both methodologies can adequately capture the effects of "remote synchronization" restoring the results obtained in (*Minati et al., 2018*), Granger causality in its unconditional form was evaluated ($F_{i \rightarrow j}$) for each couple driver ($i$) target ($j$) belonging to the ring. Here, the past history of the target node $j$ was approximated as

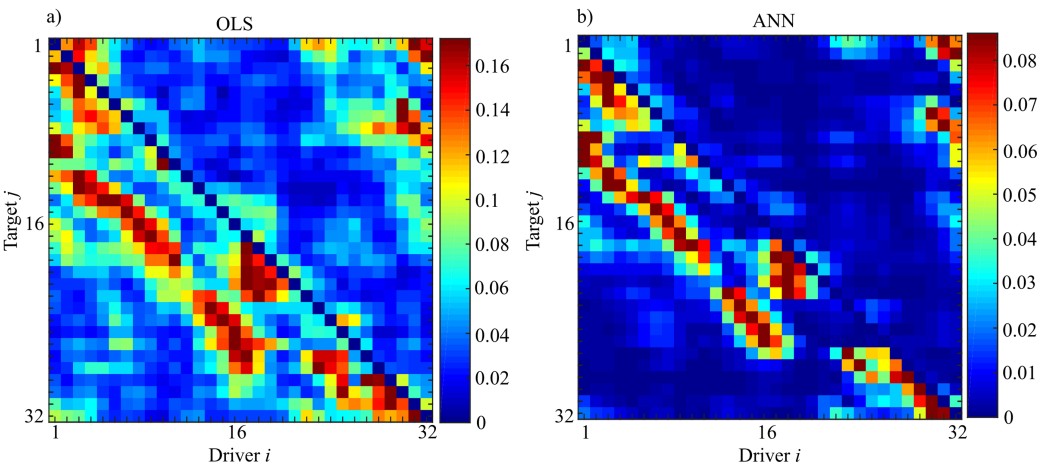

**Figure 15 Unconditional Granger Causality Analysis performed on the network of 32 chaotic oscillators ($F_{i \rightarrow j}$).** The matrices represent the analysis performed using OLS (A) and using ANNs (B) where each entry of the matrices corresponds to the strength of the causal influence from the driver $i$ towards the target $j$. The value of Sperman rank correlation coefficient ($r_s = 0.84$) reveals a strong correlation between the two different patterns ($p < 10^{-5}$).

$Y_{j,n}^p = [Y_{j,n-1}, \cdots, Y_{j,n-p}]$, i.e., with lagged components equally spaced in time. The past history of the driver node $i$ was approximated as $Y_{i,n}^p = [Y_{i,n-1}, \cdots, Y_{i,n-p}]$. In the present analyses, the model order $p$ was set to 16 with time series that were decimated firstly by a factor of 4 and subsequently by a factor 10. This process was needed in order to reduce the computational load and take into account the elimination of information storage and the propagation delays (*Minati et al., 2018*). In this condition, the ratio between the number of data samples and the number of VAR coefficients to be estimated is more or less equal to 3 ($K \approx 3$) and the partial variances needed for the evaluation of Granger causality were obtained through OLS and ANN by exploiting the theory of state-space models as described in the Methods section.

Figure 15 shows the results of the evaluation of unconditional GC ($F_{i \rightarrow j}$) performed for each couple ($i,j$) through OLS (Fig. 15A) and ANN (Fig. 15B). The estimated patterns are quite similar independently of the methodology used for estimation. The highest values of coupling estimated are linked to the previously described synchronization phenomenon: by considering a target ($j$) the coupling strength from the driver ($i$) to the considered target is very high nearby the position of the target; then decreases with the distance from the target with another peak at a distance approximately equal to 8 and finally vanishes. Another important feature is that this phenomenon is not bidirectional, but it is observable only in the direction $i \rightarrow j$ and not vice versa, as expected from the physical realization of the ring. Furthermore, the analysis of the pattern estimated through ANNs reveals more clearly the preferential synchronization clusters along the main diagonal. More in general, it is possible to observe a more sparse network when the analysis is performed through ANNs with the maximum value of observed coupling that is an order of magnitude smaller respect to the classical approach based on OLS (0.18 for OLS and 0.09 for ANNs).

The analysis of the computation time required for the estimation process, reveals a total temporal request of 28 hours ($OLS = 5.0605 \cdot 10^4$ s; $ANNs = 5.108 \cdot 10^4$ s) with the difference between the two methods ascribable to the training process of the ANN.

In order to test the degree of similarity between the two matrices, we computed the Spearman rank correlation coefficient that is a measure of the relationship between two variables when the data is in the form of rank orders. The Spearman rank correlation coefficient is in the range [−1,1] where 1 indicates complete agreement and −1 indicates complete disagreement. A value of 0 would indicate that the rankings were unrelated. Let $R_i$ be the rank of the unconditional GC evaluated through OLS and $S_i$ be the rank of the same analysis performed with ANN. Then, the rank-order correlation coefficient is defined to be the linear correlation coefficient of the ranks, namely,

$$r_s = \frac{\sum_i (R_i - \overline{R})(S_i - \overline{S})}{\sqrt{\sum_i (R_i - \overline{R})^2}\sqrt{\sum_i (S_i - \overline{S})^2}} \tag{22}$$

The significance of a nonzero value of $r_s$ is tested by computing

$$t = r_s \sqrt{\frac{N-2}{1-r_s^2}}, \tag{23}$$

which is distributed approximately as Student's distribution with N-2 degrees of freedom (*Hollander, Wolfe & Chicken, 2013*). The result of this analysis reveals a value of $r_s = 0.84$ with a *p*-value $p < 10^{-5}$ indicating a strong correspondence between the networks obtained through the two methodologies.

## DISCUSSION

### Simulation study I

The first simulation study was designed to evaluate the effects of ANN training parameters on the GC estimation process. We pointed out how the learning rate ($LR$) and the number of iterations ($N_{train}$) of the gradient descent have an impact on the training process as regards both the regression problem and the classification of significant network links (*Zhang, 2006*). The accuracy in the estimation of the regression parameters, which reflects the accuracy in the magnitude of the estimated GC, was investigated while varying the amount of data samples available for the estimation (Fig. 3). As expected, the bias of GC estimated over both null and non-null links increased in conditions of data paucity, while it was reduced increasing the number of iterations of the gradient descent. An opposite trend was observed assessing the bias along the null links for small learning rate ($LR = 10^{-5}$). This result was previously observed in the context of classification analysis (*Hoffer, Hubara & Soudry, 2017*; *Li, Wei & Ma, 2019*) and is likely due to the fact that too small learning rates can trap the ANN training process into local minima, resulting in our case in larger differences between estimated and theoretical values of the conditional GC.

On the other hand, the analysis of the accuracy in reconstructing the network structure was tested in terms of different classification parameters previously used to assess the structure of connectivity networks (*Toppi et al., 2016b*; *Antonacci et al., 2019b*; *Antonacci et al., 2020b*). The analysis (Fig. 4) showed a general improvement of the classification performance when increasing the number of data samples available and the number of iterations of the SGD-$l_1$ algorithm, and when decreasing the learning rate. These results are in line with previous studies analyzing the performance of estimators related with the concept of Granger causality (*Toppi et al., 2016a*; *Antonacci et al., 2020a*; *Antonacci et al., 2017*), and help to optimize the parameter selection for GC analysis based on ANN.

Such an optimization was performed in an objective way selecting the best combination of learning rate and number of SGD-$l_1$ iterations that minimized the overall performance parameter $S$ (Fig. 5; note that lower values of $S$ indicate better performance). Varying the parameters $N_{train}$ and $LR$ within ranges compatible with those suggested in a review of ANNs employed in classification analysis (*Zhang, 2000*), we identified the combination $LR = 10^{-3}$ and $N_{train} = 1,000$ as the most suitable for optimizing the performance of ANNs in the computation of magnitude and statistical significance of the conditional GC. Overall, our simulation results lead to the following recommendations for GC estimation based on ANNs:

- The selection of the regularization parameter $\lambda$ is crucial, and needs to be performed through objective approaches such as the use of cross-validation employed in this study. In addition, a careful selection of both the range and the number of $\lambda$ values to be tested through cross-validation is relevant; according to previous works and to the results obtained here, a range of three hundred values seems to be sufficient.
- The factors which mostly affect the computation time are the number of data samples and the number of iterations of the gradient descent ($N_{train}$). Although with a sufficient number of data samples the impact of the number of iterations does not seem to be significant, we recommend to set $N_{train} \geq 1,000$.
- Very small values of the learning rate should be avoided as they force the experimenter to increase the number of iterations of the gradient descent to escape from local minima. We suggest the combination $N_{train} = 1,000$ and $LR = 10^{-3}$ as a good compromise between accuracy and computation time.

## Simulation studies II-III

The second and the third simulation studies were designed to analyze the performance of the proposed ANN approach for GC estimation in comparison with the state-space analysis based on standard OLS estimation of the VAR model (*Barnett & Seth, 2015*) in different experimental conditions. Simulation study II has evaluated the effect of the number of data samples available (K-ratio) and the effect of the amplitude of white noise added to the original time series (SNR). Simulation study III was designed to compare the performance of the two methodologies on simulated networks with a smaller degree of sparsity with respect to the simulation study II (*Pascucci, Rubega & Plomp, 2020*). As in the

first simulation, performances were assessed separately regarding the estimation bias and the statistical significance of the conditional GC. The bias analysis revealed the expected tendency to observe a larger difference between true and estimated GC values for decreasing the $K$ ratio between amount of data samples and number of model parameters independently from the considered SNR value (Figs. 7, 9). This trend was marked for OLS-based GC estimates, confirming previous comparative studies (*Schlögl & Supp, 2006*; *Antonacci, Astolfi & Faes, 2020*), and was much less evident for ANN-based estimates, which were more stable with respect to varying $K$. Considering the worst scenario in which the number of data samples available is equal to the number of VAR coefficients to be estimated ($K = 1$), the ANN estimation still yielded acceptable results, while OLS estimation was even not possible due to the non-convergence of the DARE equation contained in the SS estimation of GC (*Antonacci et al., 2020b*). The increasing bias observed for the OLS method while approaching the condition $K = 1$ is likely related to the fact that the matrix $[y^p]^T y^p$ *(see methods)* becomes progressively closer to singularity. On the other hand, a drawback of the ANN estimator is the substantial bias exhibited by the the conditional GC computed over the non-null links even in presence of sufficient amounts of data. This could be explained in part with the penalization directly applied on the matrix of coefficients that shrinks the values towards zero, and in part to the way by which the weights of the ANN are initialized (*Scardapane & Wang, 2017*). Figure 7 highlighted a smaller effect of SNR on the bias measures in the ANN case with respect to the OLS case, which reaches values of bias very close to zero when SNR is very high ($SNR = 10^3$). This result could be explained by recalling the biased nature of regularization approaches and the tendency to counteract the effect of collinearity between regressors which may be induced by the additive noise (*James et al., 2013*).

Also the ability in reconstructing the network structure showed a tendency to decrease with the ratio $K$ between the number of data samples and model parameters (Figs. 8, 10). In terms of overall accuracy, the ANN approach outperformed the OLS one for $K \leq 3$ and $SNR = 10^3$ resulting well-applicable ($AUC \approx 0.85$) even in the challenging condition $K = 1$. We ascribe this better performance to the use of the $l_1$ regularization introduced in the training of the ANN, which helps counteracting the collinearity between regressors induced by the decrease of the number of data samples available (*Tibshirani, 1996*; *Silvey, 1969*).

As expected, the AUC parameter reported in Fig. 8 showed a tendency to decrease as a result of SNR reduction for both OLS and ANN. According to the results obtained in (*Toppi et al., 2016a*), for the OLS case the decreasing quality of the data leads to a strong increase of the estimated statistical thresholds, such that only a few connections survive to the assessment procedure. Otherwise, in the ANN case, such a trend can be explained by a wrong selection of the $\lambda$ parameter during the training procedure triggered by white noise added and is in line with a previous study in which the effect of white noise in sparse regression methods was explored (*Haufe et al., 2010*). When a more dense network is analyzed, the results of Fig. 10 showed a statistically significant difference between AUC values resulting from ANN and OLS estimation for K = [20,10] which become not statistically significant when K = 3. If compared with the results obtained in

Fig. 8 (panel c and SNR = $10^3$) a deterioration of ~ 10% can be noticed in the reconstruction of the network structure, with an AUC value of ~85% when K = 1 which becomes ~75% with a more dense network (Fig. 10C). The results here obtained are in line with those obtained in (*Antonacci et al., 2020b*) in which similar AUC trends are obtained for LASSO regression in simulated networks with a density of connected nodes equal to 50%.

When particularized to the rate of correct detection of null and non-null links, the performance under conditions of data paucity for both simulation studies differs for the two approaches, with ANN and OLS showing respectively better capability to correctly detect existing links (lower FNR) and better capability to correctly detect the absent links (lower FPR). The high rate of false negative detections exhibited by OLS when $K < 10$ is likely due to an inaccurate representation of the distribution of the GC under the null hypothesis of uncoupling, estimated empirically using surrogate time series (*Antonacci et al., 2019a*). On the other hand, the slightly higher rate of false positive detections exhibited by ANN is in line with previous findings in the context of information transfer estimation, in which the use of variable selection techniques showed few extra links, observed for different degrees of sparsity of the simulated network structure and values of $K$ (*Antonacci et al., 2020b*; *Haufe et al., 2010*; *Antonacci, Faes & Astolfi, 2020*). The latter result is also confirmed by the value of false positives obtained for ANN case in Fig. 10B in which ANN proved to be more susceptible to false positives when a denser network structure is analyzed.

Concerning the effects of SNR (Fig. 8), the decrease in the signal-to-noise ratio, regardless of the value of K-ratio considered, leads to a worsening of ~20% in the value of false negatives for both methods causing a strong reduction in the AUC value. However, ANN proved to be less affected by the reduction of signal-to-noise ratio and, even in the worst scenario of K = 1 and SNR = 0.1 the average value of AUC is above ~ 75% (Fig. 8C-purple line). These results are in line with different studies exploring the effects of SNR in Granger-based estimators in the time and frequency domain or in non-stationary regimes (*Toppi et al., 2016a*; *Pascucci, Rubega & Plomp, 2020*; *Astolfi et al., 2007*).

In sum, we provide the following remarks about the comparison between the two methods:

- If one is interested in the reconstruction of the network topology, ANNs can be used as a valid alternative to standard OLS approaches with a considerable computational cost reduction (Table 4).
- The capabilities in reconstructing the network topology of both methodologies are strongly influenced by the signal-to-noise ratio and the network density, with ANN performing better if sparse networks are considered and OLS which is more vulnerable to low SNR values.
- If one is interested in the assessment of coupling strength as measured by the GC values, ANNs are much more accurate than OLS in detecting small or zero GC values but are more biased in the detection of non-zero GC values.

- The use of ANNs with the parameter combination $N_{train} = 1,000$, $LR = 10^{-3}$ guarantees a good level of accuracy in the estimation of GC even for conditions of strong data paucity.

Two final issues should be discussed concerning the structures of simulated networks and a technical aspect related to the methodology proposed. In functional brain networks analysis, the topology of network interactions is often represented by a full AR process for which AR coefficients are non-zero at each considered lag but fade out exponentially as the lags increase. Even if the structure of the AR process here simulated was not completely full, the network structure of simulation study III was used in a previous study that approximates properties of realistic brain networks, extending beyond classical approaches with a restricted number of nodes and fixed connectivity patterns (*Pascucci, Rubega & Plomp, 2020*). In fact, with a completely full AR process it should be better to use regression analysis with a structural constraint such as group LASSO which outperforms $l_1$ regularization techniques without structural constraint (i.e., LASSO regression and the methodology here proposed) as discussed in previous works (*Mullen et al., 2015*; *Haufe et al., 2010*). Nevertheless, it is worth stressing that the formulation here introduced can be easily extended to a similar form of group sparse development inspired by the group LASSO regression, by forcing all outgoing connections from a single neuron (corresponding to a group) to be either simultaneously zero or not as reported in (*Scardapane et al., 2017*).

As a final remark, we want to emphasize that, even if the $l_1$-regularized (SGD-$l_1$) and $l_1$-constrained (LASSO) algorithms target different objective functions, their behavior could be related since the idea at the basis of their functioning is the same (*Tsuruoka, Tsujii & Ananiadou, 2009*). Nevertheless, the advantage of this type of formulation lies in the fact that it can be used indifferently with several types of loss functions (e.g., cross-entropy loss), or with different structures of the neural network designed to model non-linear relationships between input and output layers (i.e., the past states of the whole system and the present state of the target process) (*Tsuruoka, Tsujii & Ananiadou, 2009*; *Scardapane & Wang, 2017*).

## Application to physiological networks

Within the emerging field of network physiology, it is possible to analyze physiological interactions in a multivariate fashion, building complex networks whose nodes and edges represent different organ systems and their communication mechanisms (*Bashan et al., 2012*). However, identifying networks on the basis of the information exchanged between physiological signals is not a trivial task and requires the development of novel approaches (*Faes et al., 2017b*). As a main challenge is to interpret dense networks in terms of the underlying physiological mechanisms (*Faes et al., 2015*; *Porta & Faes, 2015*), the study performed here was aimed to show the usefulness of GC measures based on ANNs for the description of brain, peripheral, and brain-heart interactions in a previously studied dataset (*Zanetti et al., 2019*). The usability of the proposed approach can be inferred linking the present results to those that we obtained in recent studies where the possibility

to describe the topology of physiological networks through penalized regressions was explored (*Antonacci et al., 2020b*; *Antonacci et al., 2020a*). In particular, the very similar network topologies observed here and in (*Antonacci et al., 2020b*) using very different identification methods support the usefulness of sparse model identification approaches for the study of physiological interactions.

The analysis of the statistically significant values of the conditional GC led us to detecting specific topology structures (Fig. 11). In the study of the peripheral sub-network of cardiovascular and respiratory interactions, we confirm the results of previous works highlighting the presence of significant interaction patterns which are observed consistently across physiological states (*Zanetti et al., 2019*; *Porta et al., 2017*; *Antonacci et al., 2020b*). These patterns comprise a strong information flow between $\eta$ and $\rho$ reflecting the mechanisms of respiratory sinus arrhythmia (*Berntson, Cacioppo & Quigley, 1993*) and cardio-respiratory synchronization (*Schäfer et al., 1998*), the causal interaction $\eta \rightarrow \pi$ reflecting the physiological effect of the heart rate on stroke volume and arterial pressure which modulates the arterial pulse wave velocity (*Javorka et al., 2017*), and the causal interaction $\rho \rightarrow \pi$ reflecting the influences of breathing on the intra-thoracic pressure, blood pressure and blood flow velocity (*Drinnan, Allen & Murray, 2001*). The main effect observed when changing the physiological state was the statistically significant decrease of the in-strength index of the vascular node $\pi$ occurring with the transition from R to G (Fig. 12); physiologically, this variation can be related to a reduced efferent nervous system activity from the cardiac and respiratory centers towards the vascular system during mental stress conditions (*Antonacci et al., 2020b*; *Antonacci et al., 2020a*; *Pernice et al., 2020*). While the majority of these patterns were observed identically by OLS and ANN identification approaches, the interaction between $\rho$ and $\eta$ was detected as bidirectional using OLS and as unidirectional using ANN; the presence of unidirectional interactions $\rho \rightarrow \eta$ is physiologically more plausible with the mechanism of respiratory sinus arrhythmia (*Berntson, Cacioppo & Quigley, 1993*; *Faes, Porta & Nollo, 2015*).

As regards the analysis of the brain sub-network, we detected interaction patterns which are weaker and less consistent across physiological states. Using OLS, the total number of connections shows a tendency to decrease moving from R to M and to G. Using ANN, the brain sub-network is very sparse during R and M, and disconnected during G. The latter result is in line with our recent work in which the same dataset was analyzed through different measures of information dynamics computed through LASSO regression (*Antonacci et al., 2020b*). In such work, a different degree of disconnection was observed for the brain sub-network; given the general weakness of the connections, it is reasonable to assume that the results are influenced by the selection the regularization parameter $\lambda$ that controls the amounts of shrinkage applied to the ANN weights, as in the optimization of $\lambda$ the weaker connections have a higher probability to be discarded (*Tibshirani, 1996*; *Tibshirani & Taylor, 2012*). This confirms the importance of employing automatic strategies, such as that used in this work, for the selection the regularization parameter, in order to provide an objective quantification of the network topology. Here, the adoption of an automatic strategy led to detect a much more sparsely

connected brain subnetwork using ANN than OLS, confirming results previously reported for this type of data (*Zanetti et al., 2019*).

The regularization approach implicitly present in ANN training allowed highlighting better than standard OLS analysis the modification of the structure of brain-body interactions across the considered physiological states. Indeed, while both OLS and ANN suggest an increase of the connections between brain and body during sustained attention (condition G), the results achieved with ANN highlight the emergence of causal interactions from brain to body moving from R and M to G. The rise of these connections, directed mostly to the $\rho$ and $\eta$ nodes of the peripheral sub-network, confirms the results of previous studies about the importance of the brain oscillations for attention tasks that can be correlated with the cardiac and respiratory activity (*Tort et al., 2018*; *Kubota et al., 2001*).

## Application to chaotic electronic oscillators

The recorded time series and the master-slave unidirectional structure guarantee a higher level of stationarity and more elementary dynamics with a well known a-priori topological effect compared to physiological systems. For these reasons, it is reasonable to assume that electronic oscillators could represent a useful benchmark for testing in real settings new methods developed for the study of the interactions between dynamical systems.

The second application was therefore devised to demonstrate the validity of the proposed method, based on the combination of ANN and SS modeling, to compute GC from the output signals of a network of electronic oscillators. The analysis of the cross-correlation coefficient presented in Fig. 14 revealed the existence of a preferential synchronization effect between groups of nodes that are not directly connected via a physical link and, in particular, we found a maximum of the cross-correlation coefficient at a distance $d \approx 8$. This result is in agreement with previous analyses performed in the same ring of oscillators (*Minati, 2015a*; *Minati et al., 2018*) and with the recently introduced concept of remote synchronization which reveals mutual synchronization between pairs of locally coupled groups of nodes in a network. Thus, each group of nodes remotely synchronized is physically connected through a group of intermediary nodes more weakly synchronized with them (*Gambuzza et al., 2013*).

In order to investigate if the observed remote synchronization corresponds to "remote" information transfer, we performed unconditional GC analysis with both OLS and ANN. An inspection of Fig. 15 clearly shows the good overlap between the networks estimated with the two methodologies; this result is supported quantitatively by the analysis of the Spearman rank correlation coefficient ($r_s = 0.84$, $p < 10^{-5}$). A similar analysis was performed on the same dataset by (*Minati et al., 2018*), who used uniform embedding to approximate the history of target and driver time series as $Y_{j,n}^- = [Y_{j,n-\delta}, Y_{j,n-\tau-\delta}, \cdots, Y_{j,n-p\tau-\delta}]$, $Y_{i,n}^- = [Y_{i,n-\delta-d}, Y_{i,n-\tau-\delta-d}, \cdots, Y_{i,n-p\tau-\delta-d}]$, where the additional time lag $\delta = 0.01$ ms was added to ensure the full elimination of information storage (*Wibral et al., 2013*) and the lag $d$ was introduced to account for

propagation delays and was set searching for the minimal prediction error over the range $d \in [0,2]$.

Here, we confirm the results obtained in *Minati et al. (2018)* with a different analysis that exploits the SS representation of the VAR model and the ANN training. In particular, both methodologies can capture the dynamical activity in a ring of electronic oscillators with a well-defined complexity and stability of the network topology, since it is possible to obtain structures overlapped with those extracted performing the analysis with different methodologies already reported in the literature. From a methodological point of view, the strong overlap between the two networks can be motivated by the results of the simulation study II for which at $K = 3$ the AUC parameter, indicating the capability in the reconstruction of the network topology, showed a very small difference between the two methods. Furthermore, it is also important to note that, as an effect of the $l_1$-norm applied to the weights of the network during the training process, the maximum value of GC estimated with ANN is one order of magnitude less for ANN than OLS (*Sun et al., 2016*).

## CONCLUSIONS AND LIMITATIONS

This work documented that neural networks can be used in combination with state-space models for the identification of linear parametric models, allowing computationally reliable and accurate estimation of GC in its conditional and unconditional forms. In particular, we showed how this combined approach leads to overcoming both the decrease in accuracy reported for traditional least-squares identification when it needs to be performed in unfavorable conditions of data availability (*Schlögl & Supp, 2006*), and the problems arising in the computation of GC estimated through different regression problems (*Faes, Stramaglia & Marinazzo, 2017*). ANNs are useful in particular to assess the statistical significance of GC estimates, favoring the reconstruction of the network topology underlying the observed dataset without the need to employ time-consuming asymptotic or empirical procedures for significance assessment.

The implementation of the proposed approach for the study of physiological networks and coupled electronic oscillators documented its usefulness in practical applications, supported by the observation of interaction patterns similar to those found in previous studies where the datasets were first studied in terms of GC (*Zanetti et al., 2019*; *Minati et al., 2018*). All the findings in this work suggest that ANNs are able to detect the strongest interactions providing output patterns of information dynamics which are more straightforward and easy to interpret than those obtained with OLS.

An aspect not directly investigated in this work, that will be addressed with further studies, concerns the effect of sparsity operated by $l_1$-constrained (e.g., LASSO regression) and $l_1$-regularized (e.g., ANN here proposed) on GC measures that explicitly re-elaborate the VAR parameters. The induced sparsity in the time domain might introduce uneven shrinking of the VAR coefficients over lags which eventually causes undesired alterations in the frequency domain and this could impact the accuracy of several Granger-based estimators in the frequency domain such as Partial Directed Coherence (*Baccalá &*

*Sameshima, 2001*), Directed Transfer function (*Kamiński et al., 2001*) or Granger causality in the frequency domain (*Barnett & Seth, 2014*).

Future developments will aim at exploring the possibility of evaluating GC with non-linear ANNs trained with SGD-$l_1$ to guarantee sparseness in the estimated patterns of causality. Although $l_1$-regularized and $l_1$-constrained learning algorithms are not directly comparable due to their different objective functions, a comparison of the two approaches in term of practicality is of interest in the field of stochastic optimization (*Tsuruoka, Tsujii & Ananiadou, 2009*). Furthermore, an extensive comparison between the well-known LASSO regression and the ANN based approach here proposed, in different conditions of density of connected nodes and signal-to-noise ratio, may provide useful insights in the use of either approach (*Pagnotta, Plomp & Pascucci, 2019*; *Pascucci, Rubega & Plomp, 2020*; *Antonacci et al., 2020b*).

Given the tight relation between information dynamics and the VAR representation of Gaussian stochastic processes, future works can be envisaged to introduce ANNs for the estimation of measures of information dynamics different than the GC (*Faes et al., 2017b*; *Finn & Lizier, 2020*), computed even across multiple time scales (*Faes, Marinazzo & Stramaglia, 2017*; *Martins et al., 2020*). Moreover, this new method will easily find application even in different contexts, such as the study of dynamic information flow between stock market indices (*Scagliarini et al., 2020*), between different brain regions with Granger-based estimators (*Astolfi et al., 2007*), for time series analysis in climatology (*Faes et al., 2017a*), or for the study of gene regulatory networks (*Davidson & Levin, 2005*).

### Funding
The study was supported by Sapienza University of Rome—Progetti di Ateneo 2017 (RM11715C82606455), 2018 (RM11916B88C3E2DE), 2019 (RM11916B88C3E2DE), Progetti di Avvio alla Ricerca 2019 (AR11916B88F7079E); by Stiftelsen Promobilia, Research Project DISCLOSE; by Ministero dell'Istruzione, dell'Università e della Ricerca—PRIN 2017 (PRJ-0167), "Stochastic forecasting in complex systems", "Dipartimenti di eccellenza", PON R&I 2014-2020 AIM project (AIM1851228-2) and by BitBrain award (B2B Project 2962). There was no additional external funding received for this study. The funders had no role in study design, data collection and analysis, decision to publish, or preparation of the manuscript.

### Grant Disclosures
The following grant information was disclosed by the authors:
Sapienza University of Rome.
Progetti di Ateneo 2017: RM11715C82606455.
Progetti di Ateneo 2018: RM11916B88C3E2DE.
Progetti di Ateneo 2019: RM11916B88C3E2DE.
Progetti di Avvio alla Ricerca 2019: AR11916B88F7079E.
Ministero dell'Istruzione, dell'Università e della Ricerca—PRIN 2017: PRJ-0167.

Dipartimenti di eccellenza, PON R&I 2014-2020 AIM project: AIM1851228-2.
BitBrain award: B2B Project 2962.

## Competing Interests

The authors declare that they have no competing interests.

## Author Contributions

- Yuri Antonacci conceived and designed the experiments, performed the experiments, analyzed the data, performed the computation work, prepared figures and/or tables, authored or reviewed drafts of the paper, and approved the final draft.
- Ludovico Minati conceived and designed the experiments, performed the experiments, analyzed the data, performed the computation work, prepared figures and/or tables, authored or reviewed drafts of the paper, and approved the final draft.
- Luca Faes performed the experiments, analyzed the data, performed the computation work, prepared figures and/or tables, authored or reviewed drafts of the paper, and approved the final draft.
- Riccardo Pernice performed the experiments, analyzed the data, prepared figures and/or tables, authored or reviewed drafts of the paper, and approved the final draft.
- Giandomenico Nollo conceived and designed the experiments, performed the experiments, authored or reviewed drafts of the paper, and approved the final draft.
- Jlenia Toppi conceived and designed the experiments, authored or reviewed drafts of the paper, and approved the final draft.
- Antonio Pietrabissa conceived and designed the experiments, authored or reviewed drafts of the paper, and approved the final draft.
- Laura Astolfi conceived and designed the experiments, authored or reviewed drafts of the paper, and approved the final draft.

## Data Availability

The code necessary for the computation of Granger causality based on state-space models performed through artificial neural networks and electronic oscillator data is available at GitHub: https://github.com/YuriAntonacci/ANN-GC-Toolbox.

## Supplemental Information

Supplemental information for this article can be found online at http://dx.doi.org/10.7717/peerj-cs.429#supplemental-information.

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
