# Peer review of "Estimation of Granger causality through Artificial Neural Networks: applications to physiological systems and chaotic electronic oscillators"

_PeerJ Computer Science, doi:10.7717/peerj-cs.429_

## Round 0.1 · original submission · Major Revisions

Dear Authors, Reviewers suggest changes to your paper by solving important issues. Please revise it and return to us with revisions for further processing.

Reviewer 1 ·

Basic reporting

Work written carefully. I have no objections to the editing side of the work.

Experimental design

Well presented results.

Validity of the findings

The article is clearly written and transparent so I have no critical comments.

Additional comments

Write what are other possible practical applications of the methods presented in the paper.

Reviewer 2 ·

Basic reporting

The paper is very well written throughout. The theory and methods are both clearly explained and articulated. I enjoyed reading the paper, but I have two comments to make.

My first comment relates relates to the high level description of the method. The paper presents the method as being ANN based, but it only uses a single layer. As I understand it, this would be somewhat equivalent to an ordinary least squares, and indeed the authors seem to be getting at this point when they say:

"Thus, the described ANN is completely equivalent to a VAR model, except for the fact that the training process induces sparsity into the weight matrix W."

The sparsity constraints seem to be the most important feature here. As such, I found it a bit strange to call it an ANN since it seem to be doing linear regression with sparsity constraints. I also think that would be a bit easier to explain. I do want to emphasis, however, that this comment is only about the description. The method proposed in the paper is actually quite a nice approach.

My second comment is that description of the physiological case study could be improved. As I understood it, the entire experiment was captured in one time series whilst the subject was working through a set of different tasks. I assume that the authors have then split the time series into separate experiments, which is fine. However, if the authors have estimated the Granger Causality across the entire experiment, then the time series would hardly be stationary, which would be problematic. I don't think there is any issue here, rather I just thing the description could be improved.

Experimental design

I believe that the people who would be most interested in applying the findings from this paper will be applied scientist in biology, economics and neuroscience. Nevertheless, in my opinion, the paper is definitely within the scope of the journal since it focuses on the theoretical and technical aspects of the implementation. The work is of a high standard and is suitable for publication in this journal.

Validity of the findings

Everything looked fine to me.

Additional comments

Just so the authors can contextualise my review, my background is in the information theory and the estimation of the information-theoretic measures, i.e. I am not an expert in neural networks, so if I have misunderstood something and it is entirely different to ordinary least squares, then the authors may disregard the first of my above comments. That being said, some clarification around this point would improve the paper.

·

Basic reporting

The manuscript is well written, in a clear and adequate form.
The essential literature and background are properly described in the Introduction. Methods and results are clearly explained, and the terminology follows scientific standards.
The shared code is clean, sufficiently commented, and well organized, with sporadic comments in Italian that require an update.

Experimental design

The work fits within the aims and scopes of the journal.
The authors developed an Artificial Neural Network (ANN)-based method for estimating AR processes in the context of Granger Causality (GC) through State-Space modeling (SS). The performance and optimal settings of the ANN were investigated in a first study and compared against the classical Ordinary Least Squares (OLS) approach in a second simulation study. In two additional investigations, the authors compared ANN and OLS performance using physiological time-series and data generated by electronic oscillators.
The main goal was to propose a method for SS-GC that promotes sparsity and robustness against data paucity. Overall, most of the metrics of comparison are adequate and informative, revealing how ANNs present a valid and less time-consuming alternative to OLS in reconstructing the topology of GC interactions in simulated and real networks.
The simulations used, however, represent a very restricted scenario in which the ground truth is topologically sparse (e.g., networks with a density of 15%) and the AR process is reduced (e.g., lags with non-zero coefficients are randomly assigned to a subset of the possible 1-16 lags, if my reading is correct). This creates conditions, in simulated data, that favor sparser solutions (both in the topology and lag space). In more realistic conditions, such as in functional brain networks, the topology of network interactions often presents denser structures and full AR processes (e.g., AR coefficients that are non-zero at each considered lag, but fade out exponentially as the lag increases). It is therefore important, for the sake of application, to discuss or even test the performance of the proposed method under such circumstances.
Similarly, simulations are performed with a fixed level of SNR, but SNR is a crucial factor to assess the performance of different methods in GC, particularly when l1 regularized solutions are at stake. For this reason, the robustness to SNR variations should be evaluated as an additional factor and compared across methods.
It seems that the main advantages of ANN over OLS rely on the intrinsic l1 regularization. A more stringent comparison, to evaluate the advantage of an ANN against the pure regularization, would be to use a LASSO-penalized LS as the comparison (e.g., Pagnotta et al., 2019, Antonacci et al., 2020). The authors should at least mention this possibility.

Validity of the findings

The reported findings are useful and informative for future methodological developments and applications of GC metrics. Some limitations should be discussed. For instance, how does the sparsity regularization affect the structure of estimated AR coefficients over lags? In many applications of VAR-GC models in neuroscience, a frequency or time-frequency representation of the AR process is often preferred for interpretation (e.g., Partial Directed Coherence). Such metrics are obtained from a z-transform of the AR coefficients over lags. Imposing sparsity (both at the level of topology and lags) might introduce uneven shrinking of AR coefficients over lags, which eventually causes undesired alterations in their frequency content.
Also, using comparison metrics based on binary classification tasks hides potential differences between the two methods in their ability to estimate the (relative) weights of connections in a weighted network, an aspect that is typically relevant in many investigations. Using quantile-based thresholding criteria and AUC could be an alternative method that the authors may consider (Pascucci et al., 2020).

Additional comments

The paper introduces a very promising alternative to standard OLS. I believe that a few additional tests or discussions, which should be considered as midway between minor and major revisions, will improve the manuscript and characterize better the performance of the proposed ANN.

---

## Round 0.2 · accepted · Accept

Dear Authors,

The review process is now finished. All Reviewers are positive about your manuscript and therefore I am pleased to forward your manuscript with a positive recommendation.

Reviewer 1 ·

Basic reporting

OK

Experimental design

OK

Validity of the findings

OK

Additional comments

An interesting and well-written article I have no comments

Reviewer 2 ·

Basic reporting

No new comment.

Experimental design

No new comment.

Validity of the findings

No new comment.

Additional comments

Thank you for your efforts in integrating the suggestions. I think the paper is ready for publication.

·

Basic reporting

In line with my previous review, the manuscript is well written and clear, in standard scientific format and notation. The background literature is exhaustively covered and methods well described. I have no further concerns on basic reporting.

Experimental design

The Authors have successfully addressed my previous concerns with additional simulations and tests.

Validity of the findings

The additional discussion and the revised conclusion and limitation sections provide a final exhausitve content. I have no further concerns.